# How to Learn a Star:
# Binary Classification with Starshaped Polyhedral Sets

**Marie-Charlotte Brandenburg**
Ruhr Universität Bochum
Universitätsstr. 150, 44801 Bochum, Germany
`marie-charlotte.brandenburg@rub.de`

**Katharina Jochemko**
KTH Royal Institute of Technology
100 44 Stockholm, Sweden
`jochemko@kth.se`

## Abstract

We consider binary classification restricted to a class of continuous piecewise linear functions whose decision boundaries are (possibly nonconvex) starshaped polyhedral sets, supported on a fixed polyhedral simplicial fan. We investigate the expressivity of these function classes and describe the combinatorial and geometric structure of the loss landscape, most prominently the sublevel sets, for two loss-functions: the 0/1-loss (discrete loss) and a log-likelihood loss function. In particular, we give explicit bounds on the VC dimension of this model, and concretely describe the sublevel sets of the discrete loss as chambers in a hyperplane arrangement. For the log-likelihood loss, we give sufficient conditions for the optimum to be unique, and describe the geometry of the optimum when varying the rate parameter of the underlying exponential probability distribution.

## 1 Introduction

We study the problem of binary classification from a geometric and combinatorial perspective. Given a finite labeled data-set and a prescribed loss-function, we focus on characterizing the structure of those parameters that yield perfect classification – namely, the set of global minimizers of the loss function. More generally, we investigate the geometry and combinatorics of the entire *loss landscape* in parameter space. Understanding the geometry is central to analyze the behavior of learning algorithms, as, for example, the arrangement of critical points and the connectivity of minimizers influence optimization efficiency and generalization. Combinatorial structures, such as polyhedral decompositions, provide insights into how parameter spaces partition into regions of similar behavior. Naturally, these subdivisions interact with the subdivision into sets of classifiers which induce the same classification on the data, and is therefore intimately related to the VC dimension (the *Vapnik-Chervonenkis* dimension) of binary classifiers [Vapnik and Chervonenkis, 1971].

In order to make rigorous statements, we fix the function class used for classification to be a class which is suitable for the specific learning task. Fixing a large set of classifiers can lead to practical difficulties due to the complexity of the space of allowed classifiers, while a small set of classifiers may not be able to capture the nature of the underlying problem. A natural function class consists of those functions whose decision boundary – the geometric object separating the two classes – is the boundary of a convex polyhedron. Such function classes have been previously considered, for example, in Astorino and Gaudioso [2002], Manwani and Sastry [2010] and Kantchelian et al. [2014], where the optimal separating convex polyhedron is found through iteratively solving LPs, minimizing a logistic loss function and finding a large margin convex separator, respectively.

While polyhedral classifiers form a well-structured function class, they are also highly restrictive. In particular, the region enclosed by the decision boundary is necessarily convex, which may not always align with the structure of the underlying classification problem. Maintaining the piecewise

39th Conference on Neural Information Processing Systems (NeurIPS 2025).

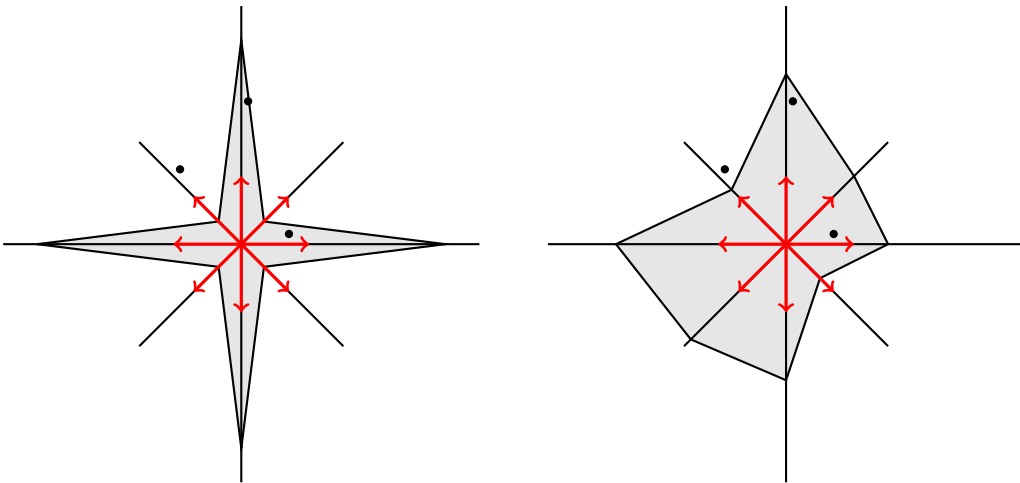

Figure 1: Two identical classifications of 3 points by different starshaped polyhedral sets, supported on the same polyhedral fan with 8 generators in $\mathbb{R}^2$.

linear nature, but allowing non-convex functions, we consider a class of piecewise linear functions whose decision boundaries are (possibly nonconvex) star-shaped polyhedral sets, supported on a fixed polyhedral fan. Fixing the polyhedral fan makes this a tractable and learnable class of functions whose space of parameters exhibits nice geometric structures as we show. Considering these function classes generalizes the approach in Cevikalp and Triggs [2017], where kites are used for solving visual object detection and multi-class discrimination.

Our family of star-shaped classifiers falls within the broader class of continuous piecewise-linear functions, and as such can be represented by a suitably structured ReLU network. However, the powerful flexibility of ReLU networks makes it challenging to enforce specific geometric properties, such as ensuring that the decision region satisfies star-convexity. Moreover, the parameter space of general ReLU neural networks admits undesirable combinatorial and geometric properties such as disconnectedness and local non-global minima even for separable data Brandenburg et al. [2024]. In contrast, by building our classifiers directly on a fixed simplicial fan, we retain the ability to model non-convex boundaries and yet maintain high control over the shape and connectivity of the decision regions.

We focus on classification with polyhedral starshaped sets with respect to two loss functions: the *0/1-loss* (or *discrete loss*) allows us to study the parameter space in a combinatorial fashion using polyhedral methods. Additionally, we consider a *log-likelihood loss function*, which is amenable to numerical optimization methods, and whose level sets carry convex-geometric structures.

## 1.1 Our contributions

In this article, we consider binary classification restricted to a class of continuous piecewise linear functions whose decision boundaries are (possibly nonconvex) starshaped polyhedral sets. More precisely, for a fixed simplicial polyhedral fan we consider the class of functions whose restrictions to each cone in the fan is linear.

**Geometry of the parameter space.** We initiate the geometric study of the space of parameters defining the classification by polyhedral starshaped sets, and show that binary classifications with this model correspond to chambers in the **data arrangement**, a hyperplane arrangement within the parameter space. Investigating the expressivity of such starshaped polyhedral classifiers, we show that the **VC dimension** equals the number of rays in the fan, quantifying how the number of linear regions of the classifier impacts sample complexity.

**Geometry of sublevel sets.** We examine the effect of the choice of the loss function explicitly on two concrete loss functions, and contrast how the geometry induced by these different losses governs optimization. We show that the sublevel sets of **false positives** and **false negatives** are starshaped sets

in the parameter space, and, in case of perfect separability, sublevel sets with respect to the **discrete loss** are starconvex sets. For the **log-likelihood loss** we show concavity, and consequently convexity holds for its superlevel sets, implying that an optimum can be found in polynomial time. We give sufficient conditions for the optimum to be unique, and describe the geometry of the optimum when varying the **rate parameter** of the underlying exponential probability distribution.

**Extended parameter spaces.**    While most of our results focus on starshaped sets with fixed origin, we also consider starshaped classifiers where we allow translation of the origin, and consider the translation vector as an extra parameter. For a fixed starshaped set and varying translation vectors, we show that the discrete loss is constant on chambers in an **arrangement of stars**, but the sublevel sets are generally no longer starconvex. While this setup combined with the log-likelihood loss does in general not lead to a convex program, we show that the log-likelihood function is piecewise concave on the underlying **data fan arrangement**.

Finally, we allow to simultaneously vary the shape of the star and the position of the origin. In this case, the sublevel sets of the discrete loss are **semialgebraic sets**, i.e., finite unions and intersections of solutions to polynomial inequalities, and we show that they are not necessarily path connected. We also explore the expressivity of this larger family of translated starshaped classifiers and show that the VC dimension is $O(d^2 \log_2(d)k \log_2(k))$ if $d$ is the dimension of the ambient space and $k$ the number of maximal cells in the fan.

## 1.2    Limitations

Throughout the article, we assume a fixed simplicial fan as given. In practical applications, an appropriate fan suitable for the specific task needs to be chosen prior to the analysis. We emphasize that this paper is a purely theoretical contribution. The presented framework has not been tested on large-scale synthetic or real-world data, only small-scale experiments as presented in the end of this article have been conducted. Developing a systematic or heuristic approach for selecting parameters such as the simplicial fan, a translation vector or the rate parameter, is beyond the scope of this work.

## 2    Description of the model

### 2.1    Polyhedral geometry and stars

We begin by introducing essential notions from polyhedral geometry, and the class of classifiers we consider. For a thorough background on polyhedral geometry we refer the reader to [Ziegler, 2012, Chapters 1-2]. Examples and visualizations of polyhedral fans are given in Appendix A

**Definition 2.1.**  A set $S \subseteq \mathbb{R}^d$ is *star-convex* with respect to a center $\mathbf{o} \in S$ if for every $\mathbf{s} \in S$, the line segment $[\mathbf{o}, \mathbf{s}] = \{\mu\mathbf{o} + (1 - \mu)\mathbf{s} : 0 \le \mu \le 1\}$ is contained in $S$. In particular, a set is *convex* it is star-convex with respect to every $\mathbf{o} \in S$.

**Definition 2.2.**  A set $C \subseteq \mathbb{R}^d$ is a *polyhedral cone* if

$$C \; = \; \left\{ \sum_{i=1}^{k} \mu_i \mathbf{v}_i : \mu_1, \ldots, \mu_k \in \mathbb{R}_{\ge 0} \right\}$$

for vectors $\mathbf{v}_1, \ldots, \mathbf{v}_k \in \mathbb{R}^d, k > 0$. The vectors $\{\mathbf{v}_i\}_{1 \le i \le k}$ are called generators of $C$. We use the notation $C = \mathrm{cone}(\mathbf{v}_1, \ldots, \mathbf{v}_k)$ for a cone with these generators. If $C$ is generated by linearly independent vectors then $C$ is called *simplicial*.

**Definition 2.3.**  A hyperplane $\{\mathbf{x} \in \mathbb{R}^d : \langle \mathbf{h}, \mathbf{x} \rangle = a\}$ is a *supporting hyperplane* of the cone $C$ if $\langle \mathbf{h}, \mathbf{v} \rangle \ge a$ for all points $\mathbf{v} \in C$. A subset $F \subseteq C$ is called a (proper) **face** if $F = C \cap H$ for some supporting hyperplane $H$.

**Definition 2.4.**  A collection $\Delta$ of polyhedral cones is called a *polyhedral fan* if the following two conditions are both satisfied.

 (i)  If $C \in \Delta$ then also every face of $C$ is in $\Delta$.

 (ii)  If $C_1, C_2 \in \Delta$ then $C_1 \cap C_2$ is a face of $C_1$ and $C_2$.

Moreover, $\Delta$ is called a *simplicial fan* if it contains only simplicial cones. It is further called *complete* if the union of all cones it contains is $\mathbb{R}^d$. A full-dimensional cone of a complete fan is a *maximal cone*. The collection of generators of all cones in the fan are called the *generators of the fan*.

Intuitively speaking, a fan is a collection of cones that fit together nicely. Examples of two well-known classes of simplicial fans, namely *kite fans* and *Coxeter fans of type B*, are given in Appendix A.1.

In the following, let $\Delta$ always be a complete, simplicial fan with generators $\{\mathbf{v}_i\}_{1 \leq i \leq n}$. Further below we will also consider affine translates of $\Delta$ consisting of translated cones of the form $C + \mathbf{t}$ where $C$ is a cone and $\mathbf{t} \in \mathbb{R}^d$ is a fixed translation vector.

**Proposition 2.5.** *For every vector* $\mathbf{a} = (a_1, \ldots, a_n) \in \mathbb{R}^n$ *there is a unique function* $f_{\mathbf{a}}^{\Delta} : \mathbb{R}^d \to \mathbb{R}$ *such that* $f_{\mathbf{a}}^{\Delta}(\mathbf{v}_i) = a_i$ *for* $1 \leq i \leq n$ *and the restriction* $f_{\mathbf{a}}^{\Delta}|_C$ *is linear for any cone* $C \in \Delta$.

Indeed, for any $\mathbf{x} \in \mathbb{R}^d$ there is a unique cone $C \in \Delta$ with generators $\mathbf{v}_{i_1} \ldots, \mathbf{v}_{i_k}$ such that $\mathbf{x} = \mu_{i_1}\mathbf{v}_{i_1} + \ldots + \mu_{i_k}\mathbf{v}_{i_k} \in C$ and $\mu_{i_j} > 0$ for all $1 \leq j \leq k$. If $C$ is a full-dimensional cone, then $k = d$ and $V_C = (\mathbf{v}_{i_1} \ldots \mathbf{v}_{i_d})$ is an invertible square matrix such that $V_C (\mu_{i_1} \ldots \mu_{i_d})^T = \mathbf{x}$, so $V_C^{-1}\mathbf{x} = (\mu_{i_1} \ldots \mu_{i_d})^T$. Define $\mu_j := 0$ for $j \in \{1, \ldots, n\} \setminus \{i_1, \ldots, i_k\}$. We write $[\mathbf{x}]^{\Delta} \in \mathbb{R}^n$ for the vector $(\mu_1, \ldots, \mu_n)$ expressing $\mathbf{x}$ as a positive linear combination of the generators of the cone of $\Delta$ that it lies in. We will sometimes simply write $[\mathbf{x}]$ when the fan $\Delta$ is clear. For exemplifying computations of $[\mathbf{x}]^{\Delta}$ when $\Delta$ is the kite fan or the Coxeter fan of type B, we refer to Appendix A.1. Since $f_{\mathbf{a}}^{\Delta}(\mathbf{v}_{i_j}) = a_{i_j}$ for $1 \leq j \leq k$, the linearity of $f_{\mathbf{a}}^{\Delta}|_C$ implies

$$f_{\mathbf{a}}^{\Delta}(\mathbf{x}) = \langle [\mathbf{x}]^{\Delta}, \mathbf{a} \rangle = \mu_{i_1}a_{i_1} + \ldots + \mu_{i_k}a_{i_k}.$$

Let $X = \{(\mathbf{x}^{(i)}, y^{(i)})\}_{i=1}^m \subset \mathbb{R}^d \times \{0, 1\}$ be a binary labeled dataset. Define the $(m \times n)$-matrix $A_X$ to be such that the $i$th row is $[\mathbf{x}^{(i)}]^{\Delta}$. Then evaluating $A_X \mathbf{a}$ results in a vector whose $i$th entry is $f_{\mathbf{a}}^{\Delta}(\mathbf{x}^{(i)})$. Observe that since $\Delta$ is simplicial, the matrix $A_X$ is *sparse* in the sense that there are at most $d$ non-zero entries in every row.

We consider the task of finding a classifier $c \colon \mathbb{R}^d \to \{0, 1\}$ that predicts $y^{(i)}$ well given $\mathbf{x}^{(i)}$. Given a complete, simplicial fan $\Delta$, we consider the set of functions

$$S^{\Delta} = \{f_{\mathbf{a}}^{\Delta} \colon \mathbb{R}^d \to \mathbb{R} \mid \mathbf{a} \in \mathbb{R}_{>0}^n\}.$$

Each function $f_{\mathbf{a}}^{\Delta}$, $\mathbf{a} > \mathbf{0}$, defines a classifier $c_{\mathbf{a}} \colon \mathbb{R}^d \to \{0, 1\}$ by setting

$$c_{\mathbf{a}}(\mathbf{x}) = \begin{cases} 0 & \text{if } f_{\mathbf{a}}^{\Delta}(\mathbf{x}) \leq 1, \\ 1 & \text{otherwise.} \end{cases}$$

The *classification* according to $c_{\mathbf{a}}$ is the vector $(c_{\mathbf{a}}(\mathbf{x}^{(1)}), \ldots, c_{\mathbf{a}}(\mathbf{x}^{(m)}))$. By slight abuse of notation we also denote the set of all classifiers $c_{\mathbf{a}}$, $\mathbf{a} \geq \mathbf{0}$ by $S^{\Delta}$. The 0-class $c_{\mathbf{a}}^{-1}(0)$ is enclosed in a star-shaped set. Indeed, it is the union of simplices with vertex sets of the form $\{0, \frac{1}{a_{i_1}}\mathbf{v}_{i_1}, \ldots, \frac{1}{a_{i_k}}\mathbf{v}_{i_k}\}$ where $\mathbf{v}_{i_1}, \ldots, \mathbf{v}_{i_k}$ are the generators of a cone $C$ of $\Delta$. We call this a *star* and denote it as $\text{star}(\mathbf{a}) = c_{\mathbf{a}}^{-1}(0)$. See Figure 1 for examples.

A data point $(\mathbf{x}^{(i)}, y^{(i)})$ has a *positive label* if $y^{(i)} = 1$ and a *negative label* if $y^{(i)} = 0$. A point $\mathbf{x}^{(i)}$ is a *false positive* with respect to $\mathbf{a}$ if $f_{\mathbf{a}}^{\Delta}(\mathbf{x}^{(i)}) > 1$ and $y^{(i)} = 0$. Similarly, the data point $\mathbf{x}^{(i)}$ is a *false negative* with respect to $\mathbf{a}$ if $f_{\mathbf{a}}^{\Delta}(\mathbf{x}^{(i)}) \leq 1$ and $y^{(i)} = 1$. We denote the number of false positives and false negatives by $\text{FP}(\mathbf{a})$ and $\text{FN}(\mathbf{a})$, respectively.

## 2.2 Loss functions

In this article, we consider minimization with respect to two distinct loss functions: the 0/1-loss and a log-likelihood loss function. For the *0/1-loss* (or *discrete loss*), we seek to minimize the number of misclassifications, counting both the false positives and false negatives, i.e.

$$\text{err}(\mathbf{a}) = \text{FP}(\mathbf{a}) + \text{FN}(\mathbf{a}). \tag{1}$$

For the *log-likelihood loss function*, let $y$ be the random variable giving the class label of the random vector $\mathbf{x} \in \mathbb{R}^d$. We approximate the probability that $\mathbf{x}$ is not in the star with the cumulative distribution function of the exponential probability distribution,

$$P(y = 1|\mathbf{x}, \Delta, \mathbf{a}) = 1 - e^{-\lambda f_{\mathbf{a}}^{\Delta}(\mathbf{x})},$$

where $\lambda > 0$ is the rate parameter of the exponential distribution. The task is to find $\mathbf{a} \in \mathbb{R}_{>0}$ that maximizes the log-likelihood function

$$
\begin{aligned}
\mathcal{L}(\mathbf{a}) &= \log \left( \prod_{i=1}^{m} P(y=1|\mathbf{x}^{(i)}, \Delta, \mathbf{a})^{y^{(i)}} \left( 1 - P(y=1|\mathbf{x}^{(i)}, \Delta, \mathbf{a}) \right)^{1-y^{(i)}} \right) \\
&= \sum_{i=1}^{m} y^{(i)} \log \left( 1 - e^{-\lambda f_{\mathbf{a}}^{\Delta}(\mathbf{x}^{(i)})} \right) + (1 - y^{(i)})(-\lambda) f_{\mathbf{a}}^{\Delta}(\mathbf{x}^{(i)}).
\end{aligned}
\tag{2}
$$

We observe that $P(y = 1|\mathbf{x}, \Delta, \mathbf{a})$ approaches $0$ when $\mathbf{x}$ approaches $0$, i.e. is in the set $\mathrm{star}(\mathbf{a})$ defined by $c_{\mathbf{a}}$, and $1$ when $\mathbf{x}$ approaches infinity, i.e. is outside the star.

Note that, in principle, one can choose to approximate $P(y = 1|\mathbf{x}, \Delta, \mathbf{a})$ with any function $F(f_{\mathbf{a}}^{\Delta}(\mathbf{x}))$ were $F$ is a cumulative distributive function on $\mathbb{R}_{\geq 0}$. The choice above will be justified by its desirable properties as shown in the following sections.

## 3 Geometry of the parameter space

In this section we study the set of optimal parameters $\mathbf{a} \in \mathbb{R}_{>0}^n$ as well as the sublevel sets of the $0/1$-loss (1) and the loss function given by the log-likelihood function (2) from a combinatorial and geometric point of view. We begin by analyzing the expressivity of the classifier, i.e., we determine the VC dimension of the set of classifiers $S^{\Delta}$. Recall that a dataset is *shattered* by a class of binary classifiers if for any possible labeling of the data there is a classifier in the class that produces the same labeling. The *VC dimension* of the class of classifiers is the maximal size of a dataset that can be shattered by the class of functions.

**Theorem 3.1.** *Let $\Delta$ be a simplicial fan with $n$ generators. Then the VC dimension of the set of classifiers $S^{\Delta}$ is equal to $n$.*

### 3.1 Geometry of the 0/1-loss

We seek to understand the geometry inside the *parameter space* $\mathbb{R}_{>0}^n = \{\mathbf{a} : \mathbf{a} > 0\}$. For an example which illustrates all definitions and results stated in this and the following subsection (Sections 3.1 and 3.2), we refer to Example A.3 in Appendix A.2.

Given an unlabeled data point $\mathbf{x}^{(i)}$, we associate the classification hyperplane

$$
H_{\mathbf{x}^{(i)}} = \{\mathbf{a} \in \mathbb{R}_{>0}^n : f_{\mathbf{a}}^{\Delta}(\mathbf{x}^{(i)}) = 1\} = \{\mathbf{a} : \langle [\mathbf{x}^{(i)}], \mathbf{a} \rangle = 1\},
$$

which separates parameters $\mathbf{a}$ inducing a classifier $c_{\mathbf{a}}$ with $c_{\mathbf{a}}(\mathbf{x}^{(i)}) = 0$ from the ones with $c_{\mathbf{a}}(\mathbf{x}^{(i)}) = 1$. These hyperplanes define the *data arrangement*

$$
\mathcal{H}_X = \bigcup_{(\mathbf{x}^{(i)}, y^{(i)}) \in X} H_{\mathbf{x}^{(i)}}.
$$

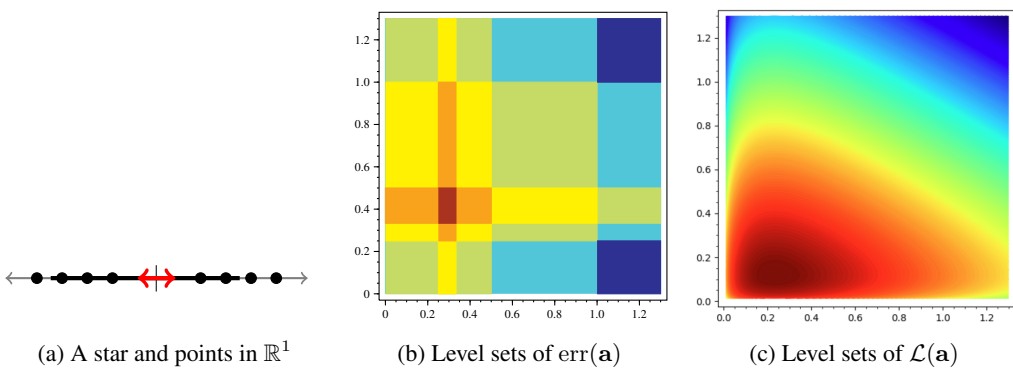

(a) A star and points in $\mathbb{R}^1$      (b) Level sets of $\mathrm{err}(\mathbf{a})$      (c) Level sets of $\mathcal{L}(\mathbf{a})$

Figure 2: An example of a 1-dimensional dataset, perfectly classified by a star supported on a fan with $n = 2$ rays, and the level sets of the two loss functions in parameter space $\mathbb{R}_{>0}^2$. This example is explained in detail in Appendix A.2.

The data arrangement subdivides the ambient space $\mathbb{R}^n_{>0}$ into (possibly empty) *half-open chambers*, i.e., subsets of the form

$$\mathcal{C}(X^0, X^1) = \bigcap_{(\mathbf{x}^{(i)}, y^{(i)}) \in X^0} \{\mathbf{a} \in \mathbb{R}^n_{>0} : f^{\Delta}_{\mathbf{a}}(\mathbf{x}^{(i)}) \leq 1\} \cap \bigcap_{(\mathbf{x}^{(i)}, y^{(i)}) \in X^1} \{\mathbf{a} \in \mathbb{R}^n_{>0} : f^{\Delta}_{\mathbf{a}}(\mathbf{x}^{(i)}) > 1\}$$

where $X^0, X^1$ is any partition of $X$. By construction, the data arrangement has the following properties.

**Proposition 3.2.** *The half-open chambers of the data arrangement are in bijection with classifications of the dataset. More precisely, each half-open chamber is the set of vectors $\mathbf{a}$ whose induced classifiers agree on the dataset.*

A direct consequence of Proposition 3.2 is that the false positives, the false negatives and thus also the discrete loss is constant on the half-open chambers. In general, since every chamber is convex and corresponds to a unique classification, the set of all parameters $\mathbf{a}$ that perfectly separate the data points is convex.

**Corollary 3.3.** *The discrete loss function $\mathrm{err}(\mathbf{a})$ is constant on the half-open chambers of the data arrangement. The parameters $\mathbf{a}$ for which $c_{\mathbf{a}}$ perfectly separate $X$ form a convex set.*

For any function $g \colon \mathbb{R}^n \to \mathbb{Z}_{\geq 0}$ and $k \in \mathbb{Z}_{\geq 0}$, the *$k$-th level set* of $g$, denoted $L(g, k)$, consists of all parameters $\mathbf{a} \in \mathbb{R}^n$ with $g(\mathbf{a}) = k$. Further, the *$k$-th sublevel set*, denoted $S(g, k)$ is defined as $S(g, k) = \bigcup_{i=0}^k L(g, i)$. In particular, all the (sub)level sets of the discrete loss function, $L(\mathrm{err}, k)$, are unions of half-open chambers. Distinguishing further between false positive and false negatives, we obtain the following geometric structure of their sublevel sets.

**Theorem 3.4.** *The sublevel sets of $\mathrm{FP}$ and $\mathrm{FN}$ are star-convex sets with star center $\mathbf{0}$ and $\infty$, respectively. That is, for all $\mathbf{a}, \mathbf{t} \in \mathbb{R}^n_{>0}$,*

*(i) $\mathrm{FP}(\mathbf{a}) \leq \mathrm{FP}(\mathbf{a} + \mathbf{t})$.*

*(ii) $\mathrm{FN}(\mathbf{a}) \geq \mathrm{FN}(\mathbf{a} + \mathbf{t})$.*

Corollary 3.3 implies that if the dataset is separable, i.e., if $L(\mathrm{err}, 0) \neq \emptyset$, then the set of perfect classifiers $L(\mathrm{err}, 0)$ is a convex set. Under the same assumption, we can make a similar statement as Theorem 3.4 about the sublevel sets of the discrete loss function.

**Theorem 3.5.** *Let $c_{\mathbf{a}}$ be a classifier that perfectly separates the dataset $X$, i.e., let $c_{\mathbf{a}} \in L(\mathrm{err}, 0)$, and let $c_{\mathbf{b}} \in L(\mathrm{err}, k)$. Then for every $\mathbf{d} \in [\mathbf{a}, \mathbf{b}]$ holds $c_{\mathbf{d}} \in S(\mathrm{err}, k)$. In particular, the sublevel sets $S(\mathrm{err}, k)$ are star-convex and connected through walls of co-dimension $1$ for every $k$.*

Theorem 3.5 shows that if the data is separable, then the sublevel sets of $\mathrm{err}$ are star-convex, but not necessarily convex in the usual sense (see also Example A.3 in Appendix A.2). Much stronger, in the case of non-separable data, sublevel sets and even the set of minimizers of the discrete loss can be disconnected. One example for this scenario is the point configuration from Example A.3, with labeling $0, 0, 1, 1, 1, 1, 0, 0$. Here, the minimum of the discrete loss-function is $4$, attained in four non-neighboring half-open chambers.

To summarize the results in this subsection, our geometric analysis reveals that sublevel sets under the 0/1-loss decompose into convex chambers within a hyperplane arrangement (Corollary 3.3), while simultaneously exhibiting star-convexity with respect to any optimal point in the chamber which minimizes the loss (Theorems 3.4 and 3.5). These theorems establish that while global optimization over the 0-1 loss is hard due to the combinatorial complexity of the overall non-convex landscape, the local landscape is well-connected. In particular, for separable data these results show that local optimization methods can between neighboring chambers along straight lines, improving in each step without getting stuck at local minima. For non-separable data, a similar behavior is still exhibited by false positives and false negatives.

## 3.2   Log-likelihood loss

In Section 3.1 we have observed that the 0/1-loss or discrete loss admits discrete geometric structures in the parameter space which are governed by an affine hyperplane arrangement. However, the discrete loss function is difficult to compute in practice. We thus propose an alternative loss-function

for practical purposes. The log-likelihood loss function turns out to be well-suited for optimization procedures due to concavity. The definitions and results stated in this subsection are exemplified in Example A.3 in Appendix A.2.

**Theorem 3.6.** *The log-likelihood function $\mathcal{L}(\mathbf{a})$ is concave. In particular, any local maximum is a global maximum.*

The computation of the maximum likelihood estimator, the maximizer of the log-likelihood loss-function, can be summarized by the following algorithm.

---
**Algorithm 1** Computation of the maximum likelihood estimator

---
**Input:** $\Delta, X = \{(\mathbf{x}^{(i)}, y^{(i)})\}_{i=1}^m, \lambda$
**Output:** $\mathbf{a}$
  1: *determine* $A_X$
  2: *solve* $\mathsf{argmax}_{\mathbf{a}>0} \sum_{i=1}^m y^{(i)} \log\left(1 - e^{-\lambda(A_X\mathbf{a})_i}\right) + (1 - y^{(i)})(-\lambda)(A_X\mathbf{a})_i$

---

Convex (and thus also concave) functions can be optimized in polynomial time. Moreover, the description in Section 2 implies that $A_X$ can be determined in polynomial time. This implies the following.

**Corollary 3.7.** *Algorithm 1 can be computed in polynomial time in the size of the input data.*

Strictly speaking, for Algorithm 1 to be a convex program, we need to consider the closed positive orthant. Any solution on the boundary corresponds to a degenerate star, where star-defining points on rays move to infinity. This degenerate case will be treated in Theorem 3.9.

Similarly to Section 3.1, we now consider the superlevel sets of the log-likelihood loss. For given $t \in \mathbb{R}$, the superlevel $S_{\mathcal{L}}(t)$ of $\mathcal{L}$ is defined as

$$S_{\mathcal{L}}(t) = \{\mathbf{a} \in \mathbb{R}_{>0}^n \colon \mathcal{L}(\mathbf{a}) \geq t\}.$$

The following is a direct consequence of the fact that $\mathcal{L}$ is concave.

**Corollary 3.8.** *The superlevel sets $S_{\mathcal{L}}(t)$ are convex sets for all $t \in \mathbb{R}$.*

We now analyze cases in which the maximum of the log-likelihood loss is unique. For this, consider the positively labeled subdataset $X_1 = \{\mathbf{x}^{(i)} \colon y^{(i)} = 1\} \subset X$ and let $A_{X_1}$ be the submatrix of $A_X$ composed of all rows $[\mathbf{x}^{(i)}]$ corresponding to $\mathbf{x}^{(i)} \in X_1$; similarly we define $A_{X_0}$ for $X_0 = X \setminus X_1$. Then we have the following sufficient condition for a unique maximum of $\mathcal{L}(\mathbf{a})$.

**Theorem 3.9.** *The log-likelihood loss function $\mathcal{L}(\mathbf{a})$ is strictly concave if the matrix $A_{X_1}$ has rank $n$. Furthermore, if also the rank of $A_{X_0}$ equals $n$, then $\mathcal{L}(\mathbf{a})$ has a unique (possibly degenerate) maximum in the closed positive orthant $\mathbb{R}_{\geq 0}^n$.*

Matrices of the form $A_X$ also appear in the study of reconstruction of polytopes with fixed facet directions from support function evaluations. Dostert and Jochemko [2023] showed that such a reconstruction of a polytope is unique if and only if $\mathrm{rank}\, A_X = n$. In other words, $A_X \in (\mathbb{R}^{n \times m}) \setminus \mathbb{V}$ where $\mathbb{V}$ is the algebraic variety encoding that $\mathrm{rank}\, A_X < n$. In particular, it is sufficient that each interior of a maximal cell of $\Delta$ contains a data point, possibly after adding minimal noise. From Theorem 3.9 together with Dostert and Jochemko [2023, Corollary 3.14] we obtain the following sufficient condition on the uniqueness of the maximum of $\mathcal{L}(\mathbf{a})$.

**Theorem 3.10.** *Let $X_0$ and $X_1$ be sets of noisy data labeled with $0$ and $1$ respectively, and such that for every maximal cell $\sigma \in \Delta$ there is at least one data point from $X_0$ and $X_1$ in the interior of $\sigma$. Then $\mathcal{L}(\mathbf{a})$ has a unique (possibly degenerate) maximum in the closed positive orthant $\mathbb{R}_{\geq 0}^n$.*

Observe that the assumptions in the preceding statement can be considered mild: Given enough data points in generic position it is reasonable to assume that the center of the star can always be chosen in a way such that the condition is satisfied.

Note that the maximizer of $\mathcal{L}$, as well as the number of false positives and false negatives also depends on the choice of the rate parameter $\lambda$. We now treat $\lambda$ as an additional variable and consider the log-likelihood function $\mathcal{L}(\lambda, \mathbf{a})$.

**Theorem 3.11.** *Let $X$ be a dataset and $\lambda_0 > 0$ be a rate parameter such that $\mathcal{L}(\lambda_0, \mathbf{a})$ has a unique maximum $\mathbf{a}^*(\lambda_0)$. Then $\mathcal{L}(\lambda, \mathbf{a})$ has a unique maximum for all $\lambda > 0$, denoted $\mathbf{a}^*(\lambda)$, and the function $\lambda \mapsto \mathbf{a}^*(\lambda)$ is a straight line inside $\mathbb{R}_{>0}^n$ approaching the origin.*

**Corollary 3.12.** *Let $X$ be a dataset such that $\mathcal{L}(\lambda, \mathbf{a})$ has a unique maximum at $\mathbf{a}^*(\lambda)$ for all $\lambda > 0$. Then*

    *(i) $\mathrm{FP}(\mathbf{a}^*(\lambda))$ is monotone decreasing in $\lambda$, and*

    *(ii) $\mathrm{FN}(\mathbf{a}^*(\lambda))$ is monotone increasing in $\lambda$ .*

Given the monotonicity of the number of false positives and false negatives in the optimal solution, it is natural to ask if the $0/1$-loss, which is given by their sum, is *convex* in the sense that it first decreases and then increases with varying $\lambda$, without any "ups" and "downs". However, this is in general not the case. A counterexample is given by Example A.3.

## 4  Geometry of the generalized parameter space

In the previous section, we have considered polyhedral fans whose cones have their apex at the origin, and varied the shapes of the stars defined on this fixed fan. In this section, we extend this framework by allowing translations. In Section 4.1 we first fix the shape of the star and only vary it by translation, whereas in Section 4.2 we investigate the space of both operations at the same time.

### 4.1  Translations of a fixed star

Recall that the star of $\mathbf{a} \in \mathbb{R}_{>0}^n$ is defined as $\mathrm{star}(\mathbf{a}) = \{\mathbf{x} \in \mathbb{R}^d : f_{\mathbf{a}}^{\Delta}(\mathbf{x}) \leq 1\}$, where $f_{\mathbf{a}}^{\Delta} = \langle [\mathbf{x}]^{\Delta}, \mathbf{a} \rangle$. Given a translation vector $\mathbf{t} \in \mathbb{R}^d$, we have

$$\mathrm{star}(\mathbf{a}) + \mathbf{t} = \{\mathbf{x} + \mathbf{t} : f_{\mathbf{a}}^{\Delta}(\mathbf{x}) \leq 1\} = \{\mathbf{x} + \mathbf{t} : \langle [\mathbf{x}]^{\Delta}, \mathbf{a} \rangle \leq 1\} = \{\mathbf{x} : \langle [\mathbf{x} - \mathbf{t}]^{\Delta}, \mathbf{a} \rangle \leq 1\} \quad (3)$$

where $[\mathbf{x} - \mathbf{t}]^{\Delta}$ captures the nonzero coefficients $\mu_i$ such that $\mathbf{x} = \mathbf{t} + \mu_{i_1} \mathbf{v}_{i_1} + \ldots + \mu_{i_k} \mathbf{v}_{i_k} \in C + \mathbf{t}$, and $C + \mathbf{t} \in \Delta + \mathbf{t}$ is the unique cone of the translated fan containing $\mathbf{x}$. On the other hand, we have

$$\mathbf{x} \in C + \mathbf{t} \iff \mathbf{t} \in \mathbf{x} - C,$$

where $\mathbf{x} - C = \{\mathbf{x} - \mathbf{c} \mid \mathbf{c} \in C\}$ is the reflected cone $-C$ translated by the vector $\mathbf{x}$. We first analyze the behavior of $[\mathbf{x} - \mathbf{t}]^{\Delta}$ when varying $\mathbf{t}$.

**Proposition 4.1.** *For any $\mathbf{x}^{(i)}$, the function $\mathbf{t} \mapsto [\mathbf{x}^{(i)} - \mathbf{t}]^{\Delta}$ is piecewise-linear, with linear pieces supported on the closed cones of the fan $\mathbf{x}^{(i)} - \Delta$. For a fixed $\mathbf{a} \in \mathbb{R}_{>0}^n$ holds*

$$\{\mathbf{t} \in \mathbb{R}^d : \langle [\mathbf{x}^{(i)} - \mathbf{t}]^{\Delta}, \mathbf{a} \rangle \leq 1\} = -\mathrm{star}(\mathbf{a}) + \mathbf{x}^{(i)}.$$

Given a fixed classifier $\mathbf{a} \in \mathbb{R}_{>0}^n$, we can ask about the nature of the *translational $0/1$-loss function*

$$\mathrm{err}_{\mathbf{a}}(\mathbf{t}) = |\{i : f_{\mathbf{a}}^{\Delta}(\mathbf{x}^{(i)} - \mathbf{t}) \leq 1, y^{(i)} = 1\}| + |\{i : f_{\mathbf{a}}^{\Delta}(\mathbf{x}^{(i)} - \mathbf{t}) > 1, y^{(i)} = 0\}|.$$

For this, we define an *arrangement of stars* $S_1, \ldots, S_n \subset \mathbb{R}^d$ to be the union of all (possibly non-convex) polyhedral sets of the form $S_1^{c_1} \cap \cdots \cap S_n^{c_n}$, where $S_j^{c_j} \in \{S_j, \mathbb{R}^d \setminus S_j\}$.

**Definition 4.2.** *Given a fixed $\mathbf{a} \in \mathbb{R}_{>0}^n$, the* data star arrangement *of a dataset $X = \{(\mathbf{x}^{(i)}, y^{(i)})\}_{i=1}^m$ is the arrangement of stars $-\mathrm{star}(\mathbf{a}) + \mathbf{x}^{(i)}$ for $i = 1, \ldots, m$.*

An example of the data star arrangement of a 2-dimensional dataset, together with the level sets of the translational $0/1$-loss is given in Example A.4 in Appendix A.3.

**Theorem 4.3.** *The translational $0/1$-loss $\mathrm{err}_{\mathbf{a}}(\mathbf{t})$ is constant on half-open cells of the data star arrangement of $X$.*

For a fixed classifier $\mathbf{a} \in \mathbb{R}_{>0}^n$ we may also consider maximizing the *translational log-likelihood function*

$$\mathcal{L}_{\mathbf{a}}(\mathbf{t}) = \sum_{i=1}^m y^{(i)} \log\left(1 - e^{-\lambda f_{\mathbf{a}}^{\Delta}(\mathbf{x}^{(i)} - \mathbf{t})}\right) + (1 - y^{(i)})(-\lambda) f_{\mathbf{a}}^{\Delta}(\mathbf{x}^{(i)} - \mathbf{t}).$$

To describe the behavior of this function, we define an *arrangement of (translated) polyhedral fans* $\Delta_1 + \mathbf{t}_1, \ldots, \Delta_k + \mathbf{t}_k$ to be the polyhedral complex consisting of all (necessarily convex) intersections of the form $(C_1 + \mathbf{t}_1) \cap (C_2 + \mathbf{t}_2) \cap \cdots \cap (C_k + \mathbf{t}_k)$ where $C_i + \mathbf{t}_i$ is a cone in $\Delta_i + \mathbf{t}_i$.

**Definition 4.4.** Given a fixed $\mathbf{a} \in \mathbb{R}^n_{>0}$, the *data fan arrangement* of a dataset $X = \{(\mathbf{x}^{(i)}, y^{(i)})\}_{i=1}^m$ is the arrangement of fans $\mathbf{x}^{(i)} - \Delta$ for $i = 1, \ldots, m$.

**Theorem 4.5.** *The translational log-likelihood function is concave on any maximal cell of the data fan arrangement. In particular, $\mathbf{t} \mapsto \mathcal{L}_{\mathbf{a}}(\mathbf{t})$ is a piecewise concave function on $\mathbb{R}^d$.*

## 4.2 Translations and transformations of the star together

In Section 3 we have considered classification by a starshaped polyhedral set in which $\mathbf{t} \in \mathbb{R}^d$ is fixed (and assumed to be the origin), and varied $\mathbf{a} \in \mathbb{R}^n_{>0}$. In Section 4.1 we have fixed $\mathbf{a} \in \mathbb{R}^n_{>0}$ and varied $\mathbf{t} \in \mathbb{R}^d$. As a final step, we will now vary the tuple $(\mathbf{a}, \mathbf{t}) \in \mathbb{R}^n_{>0} \times \mathbb{R}^d$, i.e., both parameters simultaneously, and consider the classifiers

$$c_{\mathbf{a}, \mathbf{t}}(\mathbf{x}) = \begin{cases} 0 & \text{if } f_{\mathbf{a}}^{\Delta}(\mathbf{x} - \mathbf{t}) \leq 1, \\ 1 & \text{otherwise.} \end{cases}$$

For a fixed cone $C \in \Delta$ and a data point $\mathbf{x}^{(i)}$, we consider the sets which contain those tuples $(\mathbf{a}, \mathbf{t})$ such that $\mathbf{x}^{(i)} \in C + \mathbf{t}$, and such that $\mathbf{x}^{(i)}$ lies inside or outside $\mathrm{star}(\mathbf{a}) + \mathbf{t}$, respectively:

$$\mathcal{S}^0(C, \mathbf{x}^{(i)}) = \{(\mathbf{a}, \mathbf{t}) : \mathbf{x}^{(i)} \in C + \mathbf{t}, \; \mathbf{x}^{(i)} \in \mathrm{star}(\mathbf{a}) + \mathbf{t}\},$$

$$\mathcal{S}^1(C, \mathbf{x}^{(i)}) = \{(\mathbf{a}, \mathbf{t}) : \mathbf{x}^{(i)} \in C + \mathbf{t}, \; \mathbf{x}^{(i)} \notin \mathrm{star}(\mathbf{a}) + \mathbf{t}\}.$$

**Proposition 4.6.** *The sets $\mathcal{S}^0(C, \mathbf{x}^{(i)})$ and $\mathcal{S}^1(C, \mathbf{x}^{(i)})$ are basic semialgebraic sets, i.e., finite intersections of solutions to polynomial inequalities. More precisely, each of them is the intersection of a polyhedral cone with a single quadratic inequality.*

We extend the 0/1-loss to be viewed as a function $\mathrm{err}(\mathbf{a}, \mathbf{t})$ in variables $(\mathbf{a}, \mathbf{t}) \in \mathbb{R}^n_{>0} \times \mathbb{R}^d$. In contrast to Theorem 3.5 and Theorem 4.3, the (sub)level sets in this extended product of both parameter spaces are neither polyhedral nor do they have piecewise-linear boundary, but they are semialgebraic.

**Theorem 4.7.** *The level sets and sublevel sets of the extended 0/1-loss on $\mathbb{R}^n_{>0} \times \mathbb{R}^d$ are semialgebraic sets, i.e., finite unions and intersections of solutions to polynomial inequalities. The defining polynomials have degree at most $2$.*

In the previous sections, the shape of the (sub)level sets immediately implied path-connectedness. However, in this more general framework, this property does not necessary hold.

**Theorem 4.8.** *The (sub)level sets of the extended 0/1-loss are in general not path-connected.*

We end this section by considering the expressivity of our starshaped classifiers, when translation is allowed. We give the following upper bound.

**Theorem 4.9.** *For a fixed simplicial polyhedral fan in dimension $d$ with $k$ maximal cones, the VC dimension of the class of functions $\{c_{\mathbf{a}, \mathbf{t}} : (\mathbf{a}, \mathbf{t}) \in \mathbb{R}^n_{>0} \times \mathbb{R}^d\}$ is in $O(d^2 \log_2(d) k \log_2(k))$.*

## 5 Experiments

We conducted small-scale experiments where we tested Algorithm 1, implemented in `SageMath 10.5` [The Sage Developers, 2024], on two-dimensional synthetic data. The computations were done on a MacBook Pro equipped with an M2 Pro chip and 32 GB of RAM. For comparison, we also applied several standard binary classification methods leading to convex optimization problems on the same dataset, as well as a ReLU neural network. The computation running time ranged from few seconds to one hour.

### 5.1 Data

Figure 3a illustrates 500 data points sampled from a given star-shaped region (in green) defined on eight rays. The data was generated as follows: we randomly selected the $x$- and $y$-coordinates of all points from the interval $[-1, 1]$ using a uniform distribution and discarded any resulting points $(x, y)$ lying outside the unit circle. This was done to achieve a near rotational symmetry of the data set. For each remaining point, we then checked whether it lies inside or outside the star-shaped region. The corresponding label was assigned accordingly, with a 90% probability of being correct.

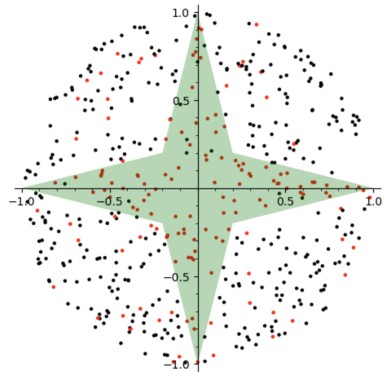

| Model | Accuracy |
|---|---|
| SVM sigmoid | 0.534 |
| SVM linear | 0.718 |
| SVM poly | 0.718 |
| Logistic reg | 0.718 |
| SVM RBF | 0.78 |
| Neural net | 0.824 |
| **Algorithm 1** | **0.852** |

(a) Synthetic data.                    (b) Results.

Figure 3: Synthetic data and accuracy of classification for tested models.

## 5.2 Results

In the first experiment, we used Algorithm 1 together with the same eight-ray fan structure to predict the labels of the synthetic data described above. We compared binary classification of our algorithm with multiple standard classification models. In the following we give a description of our results. Illustrations can be found in Appendix B

Running Algorithm 1 on the synthetic data set, the optimal value of the regularization parameter was found to be approximately $\lambda = 0.83$, yielding an accuracy of $0.852$. The resulting optimal star classifier is shown in Figure 10a. For comparison, we also tested standard implementations of SVMs (with linear, polynomial, RBF, and sigmoid kernels), logistic regression, and a ReLU neural network with two hidden layers of sizes $5$ and $2$, respectively. The SVMs with linear and polynomial kernels, as well as logistic regression, performed poorly, assigning all points to the same class, thereby achieving an accuracy of $0.718$. The SVM with a sigmoid kernel performed even worse. In contrast, the SVM with an RBF kernel and the neural network achieved better results, with accuracies of $0.78$ and $0.824$, respectively. See Figure 3b for a summary of the results.

In a further experiment, we ran Algorithm 1 on the same dataset but with different underlying fans as input. Specifically, we considered both a refinement and a coarsening of the original fan with eight rays, as depicted in Figure 11. In the case of the refined fan, the decision boundary remained almost unchanged and the accuracy improved marginally. For the coarsened fan, the shape of the decision boundary changed considerably and the accuracy became significantly worse depending on which ray was removed; see Figure 12 for an illustration if the starshaped sets. These results support the following heuristic for fan selection in two dimensions: start with a small number of rays and iteratively refine the fan by adding more rays. If the additional rays do not produce significant new dents in the boundary, they can be safely discarded.

## 6 Conclusion

This article demonstrates that polyhedral starshaped sets constitute a promising family of classifiers, striking a balance between convex polyhedral classifiers and general piecewise linear functions – the latter corresponding to the class of functions representable by ReLU neural networks. The results on VC dimensions highlight that this family remains tractable from a statistical learning perspective. This is further supported by the properties of the proposed loss functions, notably convexity and star-convexity of their (sub)level sets. Moreover, the presented framework provides a high level of flexibility, particularly due to the ability to freely choose the rate parameter $\lambda$, which enables manual adjustment of the trade-off between false positives and false negatives as needed.

The presented framework has been tested only on very few example data sets in two dimensions. It remains an open question how to optimally select the parameters, such the underlying fan, the translation vector and the rate parameter, in a manner tailored to the specific problem at hand.

## Acknowledgements

We would like to thank Maria Dostert and Mariel Supina for many fruitful discussions. We also want to thank Roland Púček for insightful conversations. MB and KJ were supported by the Wallenberg AI, Autonomous Systems and Software Program funded by the Knut and Alice Wallenberg Foundation. KJ was furthermore supported by grant nr 2018-03968 and nr 2023-04063 of the Swedish Research Council as well as the Göran Gustafsson Foundation. MB was furthermore supported by the SPP 2458 "Combinatorial Synergies", funded by the Deutsche Forschungsgemeinschaft (DFG, German Research Foundation).

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

# A Examples

## A.1 Examples of polyhedral fans (Section 2)

**Example A.1** (Kites). For $1 \leq i \leq d$, let $\mathbf{e}_i \in \mathbb{R}^d$ be the vector with $1$ in coordinate $i$ and $0$ in all other coordinates. The coordinate hyperplanes $\{\mathbf{x} \in \mathbb{R}^d : \langle \mathbf{e}_i, \mathbf{x} \rangle = 0\}$ divide $\mathbb{R}^d$ into chambers, and these chambers along with all of their faces form a simplicial fan $\Diamond$ with generators $\{\pm \mathbf{e}_i\}_{1 \leq i \leq d}$. Every star arising from this fan is necessarily convex, and such stars are called *kites*. Consider the $(d \times 2d)$-matrix $A_\Diamond = (\mathbf{e}_1, -\mathbf{e}_1, \mathbf{e}_2, -\mathbf{e}_2, \dots, \mathbf{e}_d, -\mathbf{e}_d)$. Given $\mathbf{x} \in \mathbb{R}^d$, the vector $[\mathbf{x}]^\Diamond$ is obtained as

$$[\mathbf{x}]^\Diamond = \max\left(\mathbf{0}, \mathbf{x}^T A_\Diamond\right)$$

where the maximum is taken coordinatewise. The 2-dimensional fan $\Diamond$ and the function $[\mathbf{x}]^\Diamond$ restricted to each maximal cone is depicted in Figure 4a.

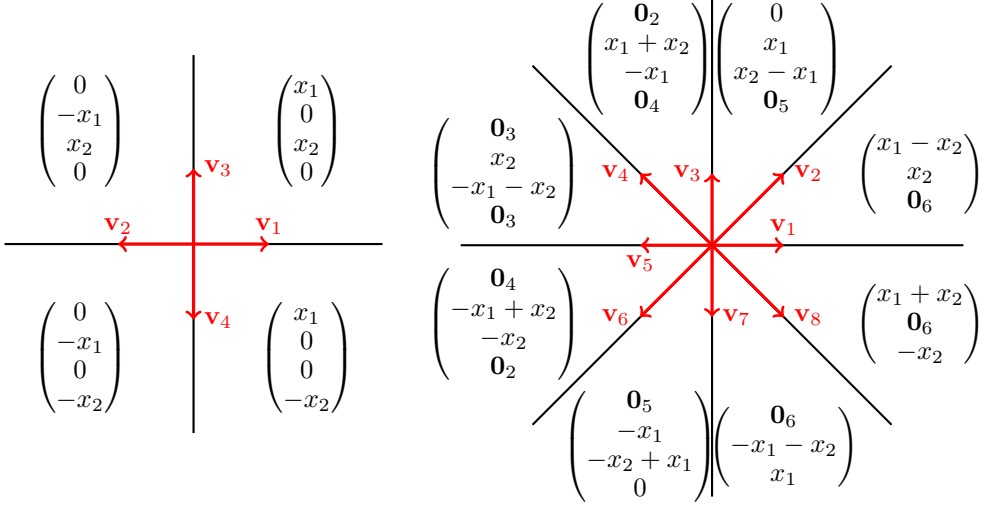

(a) The kite fan $\Diamond$.        (b) The Coxeter fan of type B.

Figure 4: The functions $[\mathbf{x}]^\Diamond$ and $[\mathbf{x}]^B$ restricted to the full-dimensional cones of the 2-dimensional fans from Examples A.1 and A.2. $\mathbf{0}_k$ denotes the $k$-dimensional 0-vector.

**Example A.2** (Type B stars). The coordinate hyperplanes along with the hyperplanes $x_i = \pm x_j$ for pairs $1 \leq i < j \leq d$ divide $\mathbb{R}^d$ into chambers, which along with their faces form a simplicial fan $B$ with generators $\{0, \pm 1\}^d \setminus \{\mathbf{0}\}$. The fan $B$ is known as the *Coxeter fan of type B*. Let $\mathbf{x} = (x_1, \dots, x_d) \in \mathbb{R}^d$ and let $\sigma : \{1, \dots, d\} \to \{1, \dots, d\}$ be a permutation such that $|x_{\sigma(1)}| \leq |x_{\sigma(2)}| \leq \cdots \leq |x_{\sigma(d)}|$. Then we have that

$$\mathbf{x} = |x_{\sigma(1)}|\mathbf{v}_1 + (|x_{\sigma(2)}| - |x_{\sigma(1)}|)\mathbf{v}_2 + \cdots + (|x_{\sigma(d)}| - |x_{\sigma(d-1)}|)\mathbf{v}_d \tag{4}$$

where $\mathbf{v}_1, \dots, \mathbf{v}_d$ generate a cone of $B$ containing $\mathbf{x}$ and

$$\mathbf{v}_i = \sum_{j=i}^d \operatorname{sgn}(x_{\sigma(i)})\mathbf{e}_{\sigma(i)}.$$

The vector $[\mathbf{x}]^B$ can be recovered from (4). The 2-dimensional fan $B$ and the function $[\mathbf{x}]^B$ restricted to each maximal cone is depicted in Figure 4b.

## A.2 Examples of the geometry of the parameter space (Section 3)

**Example A.3** (Classifications of 1-dimensional dataset). Consider the 1-dimensional polyhedral fan $\Delta$ with generators $\mathbf{v}_1 = -\mathbf{e}_1, \mathbf{v}_2 = \mathbf{e}_1$, and the 1-dimensional labeled dataset $X$ consisting of 8 distinct points

$$(\mathbf{x}^{(1)}, y^{(1)}) = (-4, 0), \ (\mathbf{x}^{(2)}, y^{(2)}) = (-3, 1), \ (\mathbf{x}^{(3)}, y^{(3)}) = (-2, 1), \ (\mathbf{x}^{(4)}, y^{(4)}) = (-1, 1),$$

$$(\mathbf{x}^{(5)}, y^{(5)}) = (1, 1), \quad (\mathbf{x}^{(6)}, y^{(6)}) = (2, 1), \quad (\mathbf{x}^{(7)}, y^{(7)}) = (3, 0), \quad (\mathbf{x}^{(8)}, y^{(8)}) = (4, 0).$$

For $z \in \mathbb{R}_{\geq 0}$ we have

$$[-z]^\Delta = \begin{pmatrix} z \\ 0 \end{pmatrix} \quad \text{and} \quad [z]^\Delta = \begin{pmatrix} 0 \\ z \end{pmatrix}.$$

The associated data arrangement is depicted in Figure 5a, and Figure 5b shows the 1-dimensional dataset together with the stars associated to the points

$$\mathbf{a}_1 = \begin{pmatrix} \frac{6}{5} \\ \frac{5}{4} \end{pmatrix}, \quad \mathbf{a}_2 = \begin{pmatrix} \frac{2}{7} \\ \frac{3}{7} \end{pmatrix}, \quad \mathbf{a}_3 = \begin{pmatrix} \frac{2}{3} \\ \frac{3}{14} \end{pmatrix}.$$

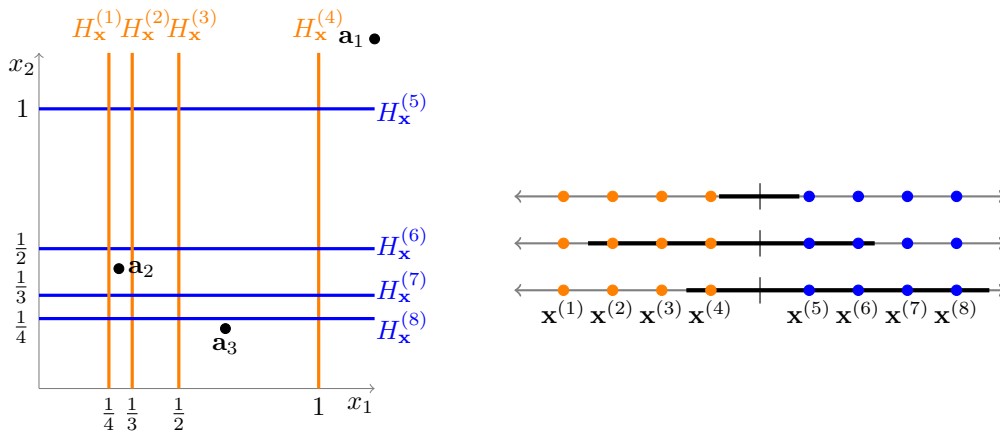

(a) Data arrangement $\mathcal{H}_X$.

(b) The dataset and the stars (thick black line) associated to $\mathbf{a}_1, \mathbf{a}_2, \mathbf{a}_3$ (top to bottom).

Figure 5: The data arrangement and dataset from Example A.3.

By Proposition 3.2, each half-open chamber of the data arrangement is the set of classification vectors $\mathbf{a}$ whose induced classifiers agree on the dataset. Thus, the number of false positives $\mathrm{FP}(\mathbf{a})$, the number of false negatives $\mathrm{FN}(\mathbf{a})$ and the discrete loss function $\mathrm{err}(\mathbf{a})$ are constant on each of the half-open chambers (cf. Corollary 3.3). Figure 6 shows the values of these functions and it can be verified that the sublevel sets of $\mathrm{FP}(\mathbf{a})$ and $\mathrm{FN}(\mathbf{a})$ are star-convex with centers $\mathbf{0}$ and $\infty$, respectively (cf. Theorem 3.4). As the data is perfectly separable, Theorem 3.5 implies that the sublevel sets of $\mathrm{err}(\mathbf{a})$ are star-convex and connected through walls of codimension 1, as depicted in Figure 6c.

Figure 7 shows the level sets of the log-likelihood loss function for the same example, for two choices of the rate parameter $\lambda$. In accordance to Corollary 3.8, the plots illustrate that the superlevel sets of

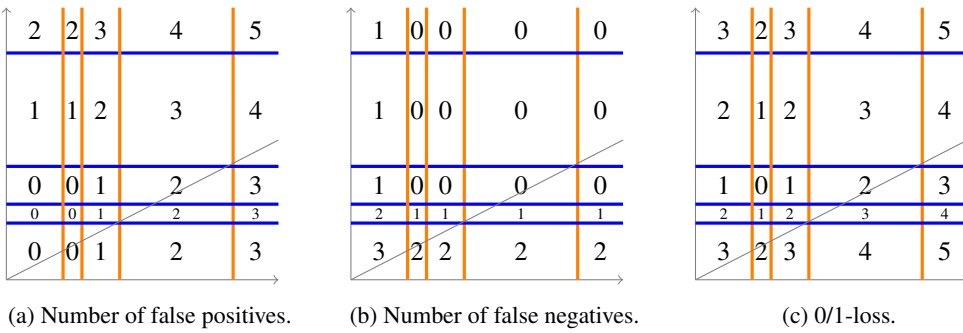

(a) Number of false positives.

(b) Number of false negatives.

(c) 0/1-loss.

Figure 6: The values of $\mathrm{FP}(\mathbf{a})$, $\mathrm{FN}(\mathbf{a})$ and $\mathrm{err}(\mathbf{a})$ on the half-open chambers of the data arrangement for the 1-dimensional dataset from Example A.3. The diagonal lines in all three images show the function $\lambda \mapsto \mathbf{a}^*(\lambda)$ defined in Section 3.2.

these functions are convex. Moreover, the rate parameter has an influence on the (unique) maximizer of these functions, but they lie on a line $\lambda \mapsto \mathbf{a}^*(\lambda)$, as shown in Theorem 3.11.

The same line is drawn in Figures 6a and 6b, illustrating that the numbers of false positives and false negative are monotone increasing and decreasing, respectively (cf. Corollary 3.12). In Figure 6c, the line crosses regions where the $0/1$-loss has values $3, 2, 3, 2, 3, 2, 3, 4$, respectively. This shows that the $0/1$-loss is not "convex" in the sense that the sequence of values of the $0/1$-loss along this line is unimodal, but goes up and down repeatedly.

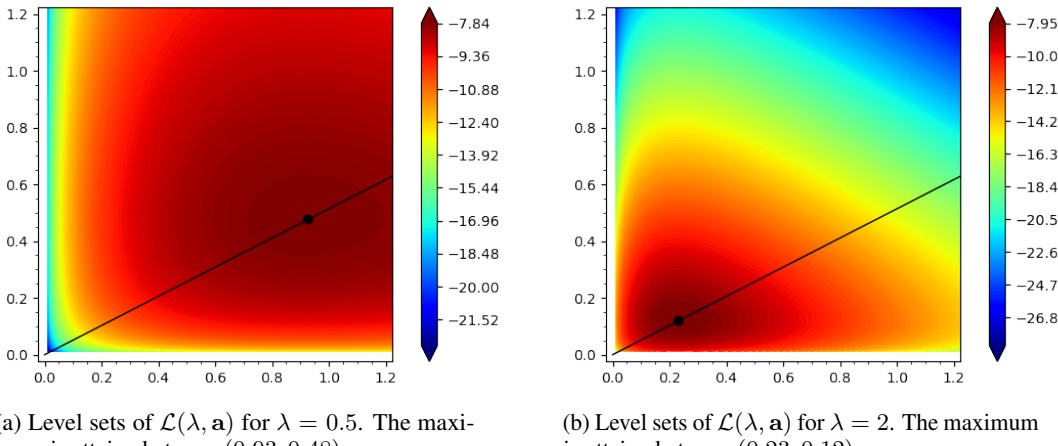

(a) Level sets of $\mathcal{L}(\lambda, \mathbf{a})$ for $\lambda = 0.5$. The maximum is attained at $\mathbf{a} = (0.93, 0.48)$.

(b) Level sets of $\mathcal{L}(\lambda, \mathbf{a})$ for $\lambda = 2$. The maximum is attained at $\mathbf{a} = (0.23, 0.12)$.

Figure 7: Level sets of log-likelihood loss functions for different choices of $\lambda$ on the dataset from Example A.3. The black dot depicts the minimum $\mathbf{a}^*(\lambda)$, lying on the line of maxima when varying the choice of $\lambda$.

### A.3 Examples of the geometry of the generalized parameter space (Section 4)

**Example A.4** (Star arrangement). Consider the 2-dimensional labeled dataset $X$ consisting of 3 distinct points

$$(\mathbf{x}^{(1)}, y^{(1)}) = (1, 1, 0), \quad (\mathbf{x}^{(2)}, y^{(2)}) = (2, 2, 1), \quad (\mathbf{x}^{(3)}, y^{(3)}) = (3, 3, 0),$$

and let $\Delta$ be the 2-dimensional Coxeter fan of type B (cf. Example A.2), which has 8 rays. We fix the star through values

$$\mathbf{a} = \left( \frac{1}{3}, 3, \frac{1}{3}, 3, \frac{1}{3}, 3, \frac{1}{3}, 3 \right).$$

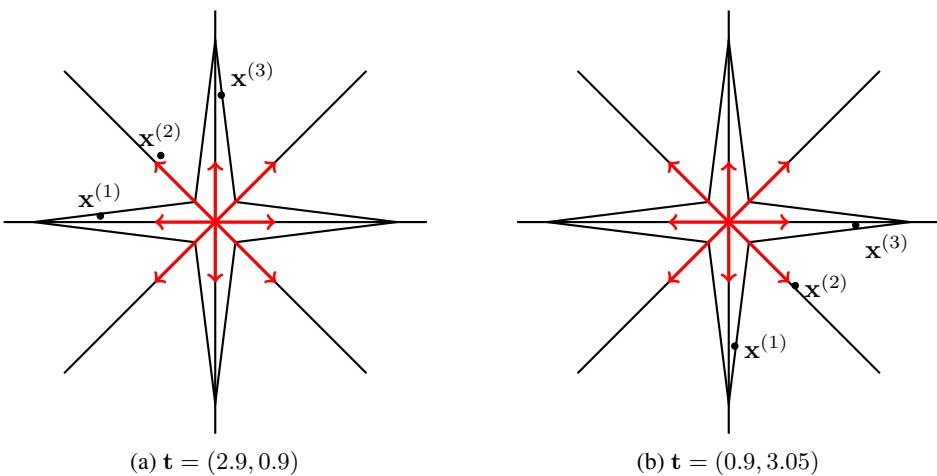

(a) $\mathbf{t} = (2.9, 0.9)$

(b) $\mathbf{t} = (0.9, 3.05)$

Figure 8: The data points from Example A.4, with fixed $\mathbf{a}$ and two perfect classifiers.

Figure 8 shows two different translations $\text{star}(\mathbf{a}) + \mathbf{t}$ that are perfect classifiers. Figure 9 shows the associated star arrangement in $\mathbb{R}^2$, and the sublevel sets of the translational 0/1-loss. In particular, in contrast to Theorem 3.5, it can be observed that the $0^{\text{th}}$ level set $L(\text{err}_{\mathbf{a}}, 0)$ consists of two full-dimensional connected components. The translated stars in Figure 8 show one example from each of the connected components.

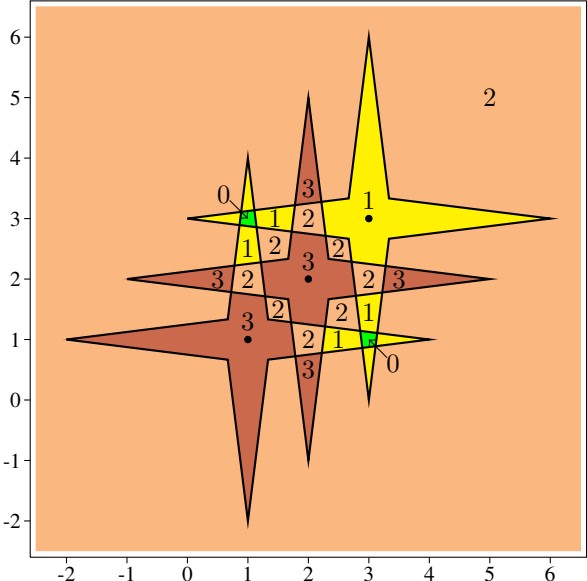

Figure 9: The star arrangement from Example A.4, and the level sets of the translational 0/1-loss function.

# B  Experiments

In this section we collect illustrations of the results of our experiments described in Section 5.

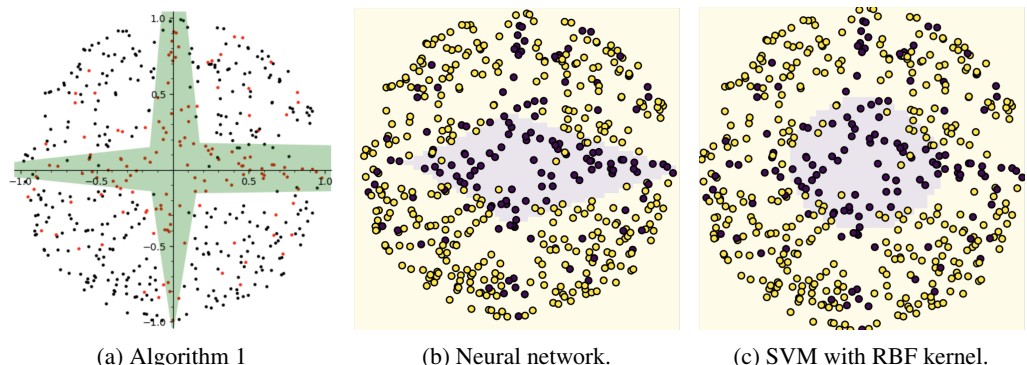

(a) Algorithm 1          (b) Neural network.          (c) SVM with RBF kernel.

Figure 10: Decision boundaries for Algorithm 1, a neural network and SVM with RBF kernel.

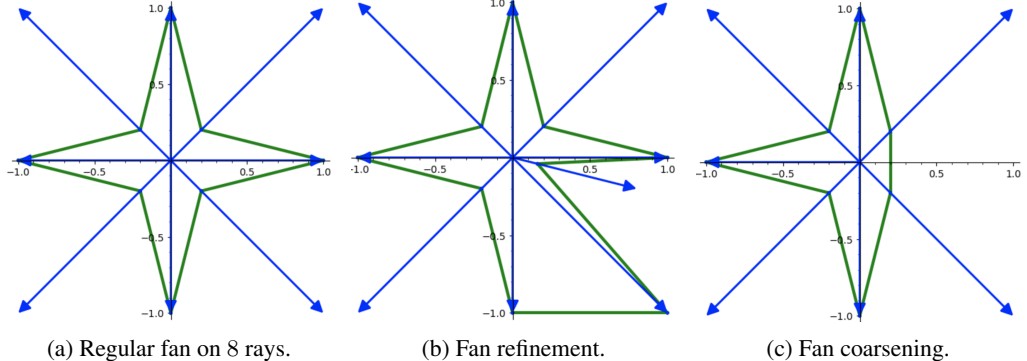

(a) Regular fan on 8 rays.   (b) Fan refinement.   (c) Fan coarsening.

Figure 11: Left: Fan on 8 rays (blue) and the supported star (green) used in the generation of the synthetic data; Center: Refinement of the fan with one extra ray, and an example of a supported star; Right: Coarsening of the fan with one ray removed, and an example of a supported star.

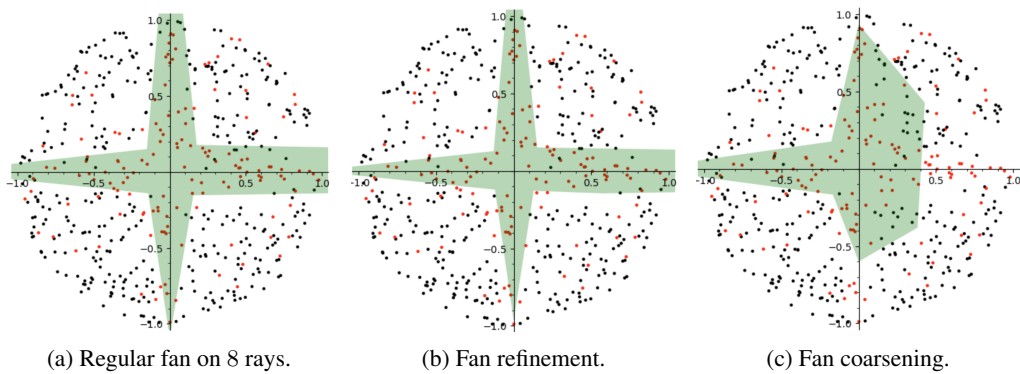

(a) Regular fan on 8 rays.   (b) Fan refinement.   (c) Fan coarsening.

Figure 12: Optimal starshaped decision boundaries for Algorithm 1, with the same underlying fan as the synthetic data (left), a refinement of the fan (center) and a coarsening of the fan (right). The underlying fans are depicted in Figure 11.

## C  Proofs

### C.1  Proof of Theorem 3.1

*Proof.* Let $\mathbf{v}_1, \ldots, \mathbf{v}_n$ be the generators of $\Delta$ and let $\ell \in \{0, 1\}^n$ be an arbitrary assignment of $0/1$-labels to these generators. Then there exists a classifier $c_{\mathbf{a}} \in S^\Delta$ that assigns the same labels to $\mathbf{v}_1, \ldots, \mathbf{v}_n$ than $\ell$, namely, for $\epsilon > 0$, we set

$$a_i = \begin{cases} 1 - \epsilon & \text{if } \ell(\mathbf{v}_i) = 0 \\ 1 + \epsilon & \text{if } \ell(\mathbf{v}_i) = 1 \,. \end{cases}$$

Then $f_{\mathbf{a}}^\Delta(\mathbf{v}_i) = a_i \geq 1$ if and only if $\ell(\mathbf{v}_i) = 1$, and thus $c_{\mathbf{a}}(\mathbf{v}_i) = \ell(\mathbf{v}_i)$ as claimed. It follows that the set of classifiers shatters the set of generators and thus the VC dimension of $S^\Delta$ is at least $n$.

To see that the VC dimension is at most $n$, we assume that there is a set $\mathbf{x}^{(1)}, \ldots, \mathbf{x}^{(n+1)} \in \mathbb{R}^d$ of $n + 1$ points that can be shattered by $S^\Delta$. By construction, for all $1 \leq i \leq n + 1$, $c_{\mathbf{a}}(\mathbf{x}^{(i)}) = 0$ if and only if $\langle [\mathbf{x}^{(i)}]^\Delta, \mathbf{a} \rangle \leq 1$. In particular, if $\mathbf{x}^{(1)}, \ldots, \mathbf{x}^{(n+1)}$ can be shattered by $S^\Delta$ then $[\mathbf{x}^{(1)}]^\Delta, \ldots, [\mathbf{x}^{(n+1)}]^\Delta$ can be shattered by the set of half-spaces of the form $\{\mathbf{x} \in \mathbb{R}^n \colon \mathbf{a}^T\mathbf{x} \leq b, a_1, \ldots, a_n \geq 0\}$. This is a subset of the set of halfspaces considered in Proposition C.1 below, and thus we obtain a contradiction by Proposition C.1. This completes the proof. □

**Proposition C.1.** *Let $\mathcal{H}$ be the set of halfspaces in $\mathbb{R}^n$ of the form*

$$\{\mathbf{x} \in \mathbb{R}^n \colon \mathbf{a}^T\mathbf{x} \leq b, a_n \geq 0\} \,.$$

Then the VC dimension of $\mathcal{H}$ equals $n$.

*Proof.* We need to show that there is no set of $n+1$ that can be shattered by $\mathcal{H}$. Let $\mathbf{y}_1, \ldots, \mathbf{y}_{n+1}$ be points in $\mathbf{R}^n$ and let $\tilde{\mathbf{y}}_1, \ldots, \tilde{\mathbf{y}}_{n+1}$ be their projections on the first $n-1$ coordinates. Then $\tilde{\mathbf{y}}_1, \ldots, \tilde{\mathbf{y}}_{n+1}$ are affinely dependent, that is, there exist $\mu_1, \ldots, \mu_n$, not all equal to zero, such that $\sum_i \mu_i = 0$ and $\sum_i \mu_i \tilde{\mathbf{y}}_i = \mathbf{0}$. Let

$$A = \sum_{i:\, \mu_i > 0} \mu_i = \sum_{i:\, \mu_i < 0} (-\mu_i).$$

Then $A > 0$ and we have that

$$\tilde{\mathbf{w}} := \frac{1}{A} \sum_{i:\, \mu_i > 0} \mu_i \tilde{\mathbf{y}}_i = \frac{1}{A} \sum_{i:\, \mu_i < i} (-\mu_i) \tilde{\mathbf{y}}_i$$

lies in the convex hull of both the sets $\{\tilde{\mathbf{y}}_i : \mu_i > 0\}$ and $\{\tilde{\mathbf{y}}_i : \mu_i < 0\}$. We consider the vectors

$$\mathbf{w}_+ = \frac{1}{A} \sum_{i:\, \mu_i > 0} \mu_i \mathbf{y}_i, \qquad \mathbf{w}_- = \frac{1}{A} \sum_{i:\, \mu_i < 0} (-\mu_i) \mathbf{y}_i.$$

Then both $\mathbf{w}_+$ and $\mathbf{w}_-$ agree with $\tilde{\mathbf{w}}$ on the first $n-1$ coordinates. W.l.o.g we may assume that $(w_-)_n \leq (w_+)_n$, that is, the last coordinate of $w_-$ is not bigger than the last coordinate of $w_+$. Then we **claim** that there exists no half-space $\{\mathbf{x} \in \mathbb{R}^n : \mathbf{a}^T \mathbf{x} \leq b, a_n \geq 0\}$ in $\mathcal{H}$ such that $\mathbf{a}^T \mathbf{y}_i \leq b$ for all $i$ such that $\mu_i > 0$ and $\mathbf{a}^T \mathbf{y}_i > b$ for all $i$ such that $\mu_i < 0$. To see this, we assume to the contrary that such a hyperplane exists. We then have

$$\mathbf{a}^T \mathbf{w}_+ = \frac{1}{A} \sum_{i:\, \mu_i > 0} \mu_i \mathbf{a}^T \mathbf{y}_i \leq \frac{1}{A} \sum_{i:\, \mu_i > 0} \mu_i b = b$$

and similarly $\mathbf{a}^T \mathbf{w}_- > b$. In particular, $\mathbf{a}^T \mathbf{w}_- > \mathbf{a}^T \mathbf{w}_+$. Since $\mathbf{w}_+$ and $\mathbf{w}_-$ agree on the first $n-1$ coordinates it therefore follows that $a_n(w_+)_n < a_n(w_-)$ and thus $\mathbf{w}_+ < \mathbf{w}_-$ since $a_n > 0$, a contradiction. Thus, no hyperplane in $\mathcal{H}$ satisfies the claim and thus $\mathcal{H}$ does not shatter any set of $n+1$ points. The VC dimension is thus at most $n$. To see that it is fact equal to $n$ we observe that the set of unit vectors $\mathbf{e}_1, \ldots, \mathbf{e}_n$ can be shattered. □

## C.2 Proof of Theorem 3.4

*Proof.* For any labeled data point $(\mathbf{x}^{(i)}, y^{(i)})$ and $\mathbf{t} \geq 0$,

$$f_{\mathbf{a}}^\Delta(\mathbf{x}^{(i)}) = \langle [\mathbf{x}^{(i)}], \mathbf{a} \rangle \leq \langle [\mathbf{x}^{(i)}], \mathbf{a} + \mathbf{t} \rangle = f_{\mathbf{a}+\mathbf{t}}^\Delta(\mathbf{x}^{(i)}).$$

In particular, if $\mathbf{x}^{(i)}$ is in the 1-class of $c_{\mathbf{a}}$ then it is also in the 1-class of $c_{\mathbf{a}+\mathbf{t}}$, so any false positive of $c_{\mathbf{a}}$ is also a false positive of $c_{\mathbf{a}+\mathbf{t}}$. This shows the first claim. The second claim follows analogously. □

## C.3 Proof of Theorem 3.5

*Proof.* Let $c_{\mathbf{a}} \in L(\mathrm{err}, 0), c_{\mathbf{b}} \in L(\mathrm{err}, k)$ and $\mathbf{d}(t) = t\mathbf{b} + (1-t)\mathbf{a}$ for $t \in [0,1]$. Let further $X_0 = \{\mathbf{x}^{(i)} : y^{(i)} = 0\}$ and $X_1 = X \setminus X_0$. Since for fixed $\mathbf{x}$ the function $f_{\mathbf{d}(t)}^\Delta(\mathbf{x}) = \langle [\mathbf{x}]^\Delta, \mathbf{d}(t) \rangle$ is linear in $t$, for $\mathbf{x} \in X_0$ holds $0 = c_{\mathbf{a}}(\mathbf{x}) \leq c_{\mathbf{d}(t)}(\mathbf{x}) \leq c_{\mathbf{b}}(\mathbf{x}) \in \{0,1\}$, and therefore $0 = \mathrm{FP}(c_{\mathbf{a}}) \leq \mathrm{FP}(c_{\mathbf{d}(t)}) \leq \mathrm{FP}(c_{\mathbf{b}})$. For $\mathbf{x} \in X_1$ holds $1 = c_{\mathbf{a}}(\mathbf{x}) \geq c_{\mathbf{d}(t)}(\mathbf{x}) \geq c_{\mathbf{b}}(\mathbf{x}) \in \{0,1\}$, and therefore $0 = \mathrm{FN}(c_{\mathbf{a}}) \leq \mathrm{FN}(c_{\mathbf{d}(t)}) \leq \mathrm{FN}(c_{\mathbf{b}})$. Since $\mathrm{err}(c_{\mathbf{d}(t)}) = \mathrm{FP}(c_{\mathbf{d}(t)}) + \mathrm{FN}(c_{\mathbf{d}(t)})$ it follows that $c_{\mathbf{d}(t)} \in S(\mathrm{err}, k)$ for all $t \in [0,1]$. This implies that the sublevel sets are star-convex. Furthermore, observe that star-convexity holds with respect to any $\mathbf{a}$ for which $c_{\mathbf{a}}$ perfectly separates the data. The set of all such $\mathbf{a}$ is a full-dimensional cell in the data arrangement. It thus follows that $S(\mathrm{err}, k)$ must consist of cells in the hyperplane arrangement that are connected through walls of codimension 1, and $L(\mathrm{err}, k)$ and $L(\mathrm{err}, k+1)$ are connected through walls of codimension 1. □

## C.4 Proof of Theorem 3.6

*Proof.* It is sufficient to show that all summands of $\mathcal{L}(\mathbf{a})$ in the expression above are concave. To that end, we observe that for given training data $(\mathbf{x}^{(i)}, y^{(i)})_{i=1,\dots,m}$, each summand $(-\lambda)f_{\mathbf{a}}^{\Delta}(\mathbf{x}^{(i)}) = -\lambda\langle[\mathbf{x}^{(i)}]^{\Delta}, \mathbf{a}\rangle$ is linear in $\mathbf{a}$ and thus concave. To see that $\log\left(1 - e^{-\lambda f_{\mathbf{a}}^{\Delta}(\mathbf{x}^{(i)})}\right)$ is concave we calculate

$$
\begin{aligned}
\frac{\partial}{\partial a_k}\frac{\partial}{\partial a_\ell}\log\left(1 - e^{-\lambda f_{\mathbf{a}}^{\Delta}(\mathbf{x}^{(i)})}\right) &= \frac{\partial}{\partial a_k}\frac{-e^{-\lambda f_{\mathbf{a}}^{\Delta}(\mathbf{x}^{(i)})}}{1 - e^{-\lambda f_{\mathbf{a}}^{\Delta}(\mathbf{x}^{(i)})}}\cdot(-\lambda)[\mathbf{x}^{(i)}]_{\ell}^{\Delta} \\
&= \lambda^2[\mathbf{x}^{(i)}]_{\ell}^{\Delta}[\mathbf{x}^{(i)}]_{k}^{\Delta}\frac{-e^{-\lambda f_{\mathbf{a}}^{\Delta}(\mathbf{x}^{(i)})}}{(1 - e^{-\lambda f_{\mathbf{a}}^{\Delta}(\mathbf{x}^{(i)})})^2}
\end{aligned}
$$

In particular, the Hessian of $\log\left(1 - e^{-\lambda f_{\mathbf{a}}^{\Delta}(\mathbf{x}^{(i)})}\right)$,

$$
\frac{-\lambda^2 e^{-\lambda f_{\mathbf{a}}^{\Delta}(\mathbf{x}^{(i)})}}{(1 - e^{-\lambda f_{\mathbf{a}}^{\Delta}(\mathbf{x}^{(i)})})^2}\left([\mathbf{x}^{(i)}]^{\Delta}\right)^T[\mathbf{x}^{(i)}]^{\Delta}
$$

is negative semi-definite and thus the likelihood function $\mathcal{L}(\mathbf{a})$ is concave. $\square$

## C.5 Proof if Theorem 3.9

*Proof.* From the proof of Theorem 3.6 we see that the Hessian of $\mathcal{L}(\mathbf{a})$ is a negative linear combination of the rank-1 matrices $\left([\mathbf{x}^{(i)}]^{\Delta}\right)^T[\mathbf{x}^{(i)}]^{\Delta}$ for $\mathbf{x}^{(i)} \in X_1$, that is, $\operatorname{Hess}\mathcal{L}(\mathbf{a}) = \sum_i \lambda_i\left([\mathbf{x}^{(i)}]^{\Delta}\right)^T[\mathbf{x}^{(i)}]^{\Delta}$ for some $\lambda_i < 0$ where the sum is over all $i$ such that $\mathbf{x}^{(i)} \in X_1$. Now let $\mathbf{v}$ be an eigenvector of $\operatorname{Hess}\mathcal{L}(A)$ with eigenvalue $\mu$. Since $A_{X_1}$ has rank $n$, there exists an $\mathbf{x}^{(i_0)} \in X_1$ with $\langle[\mathbf{x}^{(i_0)}]^T, \mathbf{v}\rangle \neq 0$. Then

$$
\mu\|\mathbf{v}\|^2 = \mathbf{v}^T\operatorname{Hess}\mathcal{L}(A)\mathbf{v} = \sum_i\lambda_i(\langle[\mathbf{x}^{(i)}]^T, \mathbf{v}\rangle)^2 \leq \lambda_{i_0}(\langle[\mathbf{x}^{(i_0)}]^T, \mathbf{v}\rangle)^2 < 0.
$$

It follows that $\mu \neq 0$ and thus $\mu < 0$. Thus, $\operatorname{Hess}\mathcal{L}(A)$ is negative definite and thus $\mathcal{L}(A)$ is strictly concave. Since the parameter space $\mathbb{R}_{>0}^n = \{\mathbf{a} : \mathbf{a} > 0\}$ is convex, $\operatorname{Hess}\mathcal{L}(A)$ has a unique maximum on the extended positive orthant $(\mathbb{R}_{\geq 0} \cup \{\infty\})^n$. If furthermore the rank of $A_{X_0}$ is $n$ then for each $j \in \{1, \dots, n\}$ there exists an $x^{(i)} \in X_0$ such that $[\mathbf{x}^{(i)}]_j > 0$. Therefore, we see that $\mathcal{L}(\mathbf{a}) \to -\infty$ whenever $a_j \to \infty$. Thus, the unique maximum must be attained in the closed positive orthant $(\mathbb{R}_{\geq 0})^n$. $\square$

## C.6 Proof of Theorem 3.11

*Proof.* We observe that for all $t > 0$ holds $\mathcal{L}(t\lambda_0, \mathbf{a}) = \mathcal{L}(\lambda_0, t\mathbf{a})$. Therefore, for fixed $t > 0$ holds

$$
\mathcal{L}(t\lambda_0, \mathbf{a}) = \mathcal{L}(\lambda_0, t\mathbf{a}) \leq \mathcal{L}(\lambda_0, \mathbf{a}^*(\lambda_0)) = \mathcal{L}(t\lambda_0, 1/t \cdot \mathbf{a}^*(\lambda_0))
$$

for all $\mathbf{a} \in \mathbb{R}_{>0}^n$. Thus, $\mathbf{a}_{t\lambda_0} = 1/t \cdot \mathbf{a}^*(\lambda_0)$ is the unique maximum of $\mathcal{L}(t\lambda_0, \mathbf{a})$, and all maxima lie on the ray $\{\mathbf{a}^*(\lambda) : \lambda > 0\} = \{\mathbf{a}^*(t\lambda_0) : t > 0\} = \{\frac{1}{t}\mathbf{a}^*(\lambda_0) : t > 0\}$. $\square$

## C.7 Proof of Corollary 3.12

*Proof.* From the proof of Theorem 3.11 we see that for any $0 < \lambda < \lambda'$ holds $\mathbf{a}^*(\lambda') = \lambda/\lambda' \cdot \mathbf{a}^*(\lambda) < \mathbf{a}^*(\lambda)$. Since $\mathbf{a}^*(\lambda) \in \mathbb{R}_{>0}^n$ both claims follow from Theorem 3.4. $\square$

## C.8 Proof of Proposition 4.1

*Proof.* We begin with the first statement. First, let $C = \operatorname{cone}(\mathbf{v}_{i_1}, \dots, \mathbf{v}_{i_d})$ be a full-dimensional cone of $\Delta$ such that $\mathbf{t}$ is contained in the interior of the cone $\mathbf{x}^{(i)} - C$ of the fan $\mathbf{x}^{(i)} - \Delta$. Equivalently, $\mathbf{x}^{(i)} - \mathbf{t} \in C$. With $V_C = (\mathbf{v}_{i_1} \dots \mathbf{v}_{i_d})$, we can thus compute $[\mathbf{x}^{(i)} - \mathbf{t}]^{\Delta} = V_C^{-1}(\mathbf{x}^{(i)} - \mathbf{t})$, which is a linear function in $\mathbf{t}$. The statement for lower-dimensional cones follows by taking limits. The second statement follows from substitution of the variables $\mathbf{x} \mapsto -\mathbf{t}$ and $\mathbf{t} \mapsto \mathbf{x}^{(i)}$ in (3). $\square$

## C.9 Proof of Theorem 4.3

*Proof.* Proposition 4.1 implies that

$$\mathrm{err}_{\mathbf{a}}(\mathbf{t}) = |\{i : \mathbf{t} \in \mathbf{x}^{(i)} - \mathrm{star}(\mathbf{a}), y^{(i)} = 1\}| + |\{i : \mathbf{t} \notin \mathbf{x}^{(i)} - \mathrm{star}(\mathbf{a}), y^{(i)} = 0\}|,$$

and is thus constant on each cell of the data star arrangement. $\square$

## C.10 Proof of Theorem 4.5

*Proof.* By Proposition 4.1, for any $i$, $\mathbf{t} \mapsto f_{\mathbf{a}}^{\Delta}(\mathbf{x}^{(i)} - \mathbf{t}) = \langle \mathbf{a}, [\mathbf{x}^{(i)} - \mathbf{t}] \rangle$ is linear in $\mathbf{t}$ on every maximal cell $F$ of the fan arrangement of $\mathbf{x}^{(i)} - \Delta$, $i = 1, \ldots, m$. Let $g_F(\mathbf{t}) = \langle \mathbf{a}, [\mathbf{x}^{(i)} - \mathbf{t}] \rangle$ be this linear function. Then the Hessian of $\mathcal{L}_{\mathbf{a}}(\mathbf{t})$ is the sum of matrices of the form

$$\frac{\partial}{\partial t_{\ell}} \frac{\partial}{\partial t_j} \log(1 - e^{-\lambda g_F(t)}) = \frac{-\lambda^2 e^{-\lambda g_F(t)}}{(1 - e^{-\lambda g_F(t)})^2} \frac{\partial}{\partial t_{\ell}} g_F(t) \frac{\partial}{\partial t_j} g_F(t).$$

Since these are negative semi-definite, this proofs the claim. $\square$

## C.11 Proof of Proposition 4.6

*Proof.* First note that

$$\mathcal{S}^0(C, \mathbf{x}^{(i)}) = \{(\mathbf{a}, \mathbf{t}) : \mathbf{x}^{(i)} \in C + \mathbf{t}\} \cap \{(\mathbf{a}, \mathbf{t}) : f_{\mathbf{a}}^{\Delta}(\mathbf{x}^{(i)} - \mathbf{t}) \leq 1\}.$$

The first set equals $\mathbb{R}_{>0}^n \times (-C + \mathbf{x}^{(i)})$. Restricted to $(\mathbf{a}, \mathbf{t})$ such that $\mathbf{t} \in \mathbf{x}^{(i)} - C$, the function $\mathbf{t} \mapsto [\mathbf{x}^{(i)} - \mathbf{t}]$ is a linear map by Proposition 4.1, so the expression $f_{\mathbf{a}}^{\Delta}(\mathbf{x}^{(i)} - \mathbf{t}) = \langle [\mathbf{x}^{(i)} - \mathbf{t}]^{\Delta}, \mathbf{a} \rangle$ is a quadratic polynomial in variables $t_1, \ldots, t_d, a_1, \ldots, a_n$. Therefore, $\mathcal{S}^0(C, \mathbf{x}^{(i)})$ it the intersection of solutions to the linear inequalities defining the polyhedral cone $\mathbb{R}_{>0}^n \times (-C + \mathbf{x}^{(i)})$, and the quadratic inequality $\langle [\mathbf{x}^{(i)} - \mathbf{t}]^{\Delta}, \mathbf{a} \rangle \leq 1$. Similarly, $\mathcal{S}^1(C, \mathbf{x}^{(i)})$ is the intersection of $\mathbb{R}_{>0}^n \times (-C + \mathbf{x}^{(i)})$ with the set of solutions to the inequality $\langle [\mathbf{x}^{(i)} - \mathbf{t}]^{\Delta}, \mathbf{a} \rangle > 1$. $\square$

## C.12 Proof of Theorem 4.7

*Proof.* We consider the subdivision of $\mathbb{R}_{>0}^n \times \mathbb{R}^d$ into sets

$$\bigcap_{C \in \Delta} \bigcap_{i=1}^{m} \mathcal{S}^{b(C,i)}(C, \mathbf{x}^{(i)}),$$

where we range over all possible $b(C, i) \in \{0, 1\}$ for all $C \in \Delta, i \in \{1, \ldots, m\}$. By Proposition 4.6, each $\mathcal{S}^{b(C,i)}(C, \mathbf{x}^{(i)})$ is basic semialgebraic with defining polynomials of degree at most 2, and hence the same holds for the above finite intersection. By construction, the extended 0/1-loss is constant on each $\bigcap_{C \in \Delta} \bigcap_{i=1}^{m} \mathcal{S}^{b(C,i)}(C, \mathbf{x}^{(i)})$. Fix $k \in \mathbb{Z}_{\geq 0}$. Then the $k^{\text{th}}$ level set $L(\mathrm{err}, k)$ of the extended 0/1-loss is the (finite) union over all sets $\bigcap_{C \in \Delta} \bigcap_{i=1}^{m} \mathcal{S}^{b(C,i)}(C, \mathbf{x}^{(i)})$ on which the extended 0/1-loss is equal to $k$. Thus, the level set is a semialgebraic set. Since sublevel sets are finite unions of level sets, the same holds for sublevel sets. $\square$

## C.13 Proof of Theorem 4.8

*Proof.* We show this statement by giving an example of a data-set with a disconnected level set $L(\mathrm{err}, 0)$. For this, we continue with Example A.4, a configuration of 3 data points in $\mathbb{R}^2$. The 2-dimensional Coxeter fan of type B has 8 rays and 8 maximal cones. Thus, the parameter space $\mathbb{R}_{>0}^8 \times \mathbb{R}^2$ is subdivided into cells of the form $\bigcap_{C \in \Delta} \bigcap_{i \in \{1,2,3\}} \mathcal{S}^{b(C,i)}(C, \mathbf{x}^{(i)})$, many of these are empty or lower dimensional. The extended 0/1-loss attains the value 0 on 16 maximal cells. To describe them, we use the indexing of the rays as depicted in Figure 4b, and denote $C_i = \mathrm{cone}(\mathbf{v}_i, \mathbf{v}_{i+1})$ for $i = 1, \ldots, 7$, $C_8 = \mathrm{cone}(\mathbf{v}_8, \mathbf{v}_1)$. One example of a valid configuration is a tuple $(\mathbf{a}, \mathbf{t})$ such that $\mathbf{x}^{(1)} \in (\mathrm{star}(\mathbf{a}) + \mathbf{t}) \cap (C_4 + \mathbf{t})$, $\mathbf{x}^{(2)} \in (\mathbb{R}^2 \setminus (\mathrm{star}(\mathbf{a}) + \mathbf{t})) \cap (C_3 + \mathbf{t})$ and $\mathbf{x}^{(3)} \in (\mathrm{star}(\mathbf{a}) + \mathbf{t}) \cap (C_2 + \mathbf{t})$, as depicted in Figure 8a. The set of $(\mathbf{a}, \mathbf{t})$ satisfying these conditions

is $\mathcal{S}^0(C_4, \mathbf{x}^{(1)}) \cap \mathcal{S}^1(C_3, \mathbf{x}^{(2)}) \cap \mathcal{S}^0(C_2, \mathbf{x}^{(3)})$. In total, the set of perfect classifiers, i.e., the $0^{\text{th}}$ level set is the union of the following 16 nonempty cells:

$$\mathcal{S}^0(C_4, \mathbf{x}^{(1)}) \cap \mathcal{S}^1(C_3, \mathbf{x}^{(2)}) \cap \mathcal{S}^0(C_2, \mathbf{x}^{(3)}), \quad \mathcal{S}^0(C_6, \mathbf{x}^{(1)}) \cap \mathcal{S}^1(C_7, \mathbf{x}^{(2)}) \cap \mathcal{S}^0(C_8, \mathbf{x}^{(3)}),$$

$$\mathcal{S}^0(C_4, \mathbf{x}^{(1)}) \cap \mathcal{S}^1(C_3, \mathbf{x}^{(2)}) \cap \mathcal{S}^0(C_3, \mathbf{x}^{(3)}), \quad \mathcal{S}^0(C_6, \mathbf{x}^{(1)}) \cap \mathcal{S}^1(C_7, \mathbf{x}^{(2)}) \cap \mathcal{S}^0(C_1, \mathbf{x}^{(3)}),$$

$$\mathcal{S}^0(C_4, \mathbf{x}^{(1)}) \cap \mathcal{S}^1(C_4, \mathbf{x}^{(2)}) \cap \mathcal{S}^0(C_2, \mathbf{x}^{(3)}), \quad \mathcal{S}^0(C_6, \mathbf{x}^{(1)}) \cap \mathcal{S}^1(C_8, \mathbf{x}^{(2)}) \cap \mathcal{S}^0(C_8, \mathbf{x}^{(3)}),$$

$$\mathcal{S}^0(C_4, \mathbf{x}^{(1)}) \cap \mathcal{S}^1(C_4, \mathbf{x}^{(2)}) \cap \mathcal{S}^0(C_3, \mathbf{x}^{(3)}), \quad \mathcal{S}^0(C_6, \mathbf{x}^{(1)}) \cap \mathcal{S}^1(C_8, \mathbf{x}^{(2)}) \cap \mathcal{S}^0(C_1, \mathbf{x}^{(3)}),$$

$$\mathcal{S}^0(C_5, \mathbf{x}^{(1)}) \cap \mathcal{S}^1(C_3, \mathbf{x}^{(2)}) \cap \mathcal{S}^0(C_2, \mathbf{x}^{(3)}), \quad \mathcal{S}^0(C_7, \mathbf{x}^{(1)}) \cap \mathcal{S}^1(C_7, \mathbf{x}^{(2)}) \cap \mathcal{S}^0(C_8, \mathbf{x}^{(3)}),$$

$$\mathcal{S}^0(C_5, \mathbf{x}^{(1)}) \cap \mathcal{S}^1(C_3, \mathbf{x}^{(2)}) \cap \mathcal{S}^0(C_3, \mathbf{x}^{(3)}), \quad \mathcal{S}^0(C_7, \mathbf{x}^{(1)}) \cap \mathcal{S}^1(C_7, \mathbf{x}^{(2)}) \cap \mathcal{S}^0(C_1, \mathbf{x}^{(3)}),$$

$$\mathcal{S}^0(C_5, \mathbf{x}^{(1)}) \cap \mathcal{S}^1(C_4, \mathbf{x}^{(2)}) \cap \mathcal{S}^0(C_2, \mathbf{x}^{(3)}), \quad \mathcal{S}^0(C_7, \mathbf{x}^{(1)}) \cap \mathcal{S}^1(C_8, \mathbf{x}^{(2)}) \cap \mathcal{S}^0(C_8, \mathbf{x}^{(3)}),$$

$$\mathcal{S}^0(C_5, \mathbf{x}^{(1)}) \cap \mathcal{S}^1(C_4, \mathbf{x}^{(2)}) \cap \mathcal{S}^0(C_3, \mathbf{x}^{(3)}), \quad \mathcal{S}^0(C_7, \mathbf{x}^{(1)}) \cap \mathcal{S}^1(C_8, \mathbf{x}^{(2)}) \cap \mathcal{S}^0(C_1, \mathbf{x}^{(3)}).$$

It can be checked that the union of the sets in each of the above columns is path connected. However, there is no path from a point in the left column to a point in the right column. If these sets were path connected, then there was a path from the configuration depicted in Figure 8a to the configuration in Figure 8b by a continuous translation of the star and by shifting the points $\frac{1}{a_i}\mathbf{v}_i$ along the rays continuously. But since $\mathbf{x}^{(2)}$ lies in the convex hull of $\mathbf{x}^{(1)}$ and $\mathbf{x}^{(3)}$ and the line segment through any of these point is parallel to the rays $\mathbf{v}_2, \mathbf{v}_6$, any such continuous transformation necessarily increases the number of mistakes at a certain point. $\qquad\square$

### C.14 Proof of Theorem 4.9

*Proof.* By Kupavskii [2020], the set of all simplices has VC dimension $O(d^2 \log_2(d))$. Any star-shaped classifier $c_{\mathbf{a},\mathbf{t}}$ is a union of $k$ simplices. Thus, the result follows from Blumer et al. [1989]. $\quad\square$

