# OpenReview forum: "How to Learn a Star: Binary Classification with Starshaped Polyhedral Sets"
_NeurIPS.cc/2025/Conference — NeurIPS 2025 poster_

### Official Review · Reviewer_cg74 · 2025-06-22

**Clarity:** 2
**Significance:** 2
**Originality:** 4
**Rating:** 5
**Confidence:** 2

**Summary:**

The paper studies a class of piecewise linear binary classifiers whose decision boundaries are starshaped polyhedral sets defined with respect to some fixed simplicial fan.
The authors show two main results.
First, they show that the VC-dimension of the class equals the number of generators of the simplicial .
Second, they analyze the loss landscape of the hypothesis class under the exponential loss, i.e., the cross-entropy loss, and the 0-1 loss.
In particular, they show that the exponential loss is a concave function in the parameter space, enbabling efficient learning and optimization.
They also show that sublevel sets under the 0/1-loss are structured as chambers in a hyperplane arrangement.

**Questions:**

Can you elaborate the significance of Corollary 3.3, Theorem 3.4 and Theorem 3.5?
I understand that they are interesting geometric properties of the hypothesis class. But does it have implication on the advantages or disadvantages of using the 0-1 loss function for optimizing the hypothesis class?

**Ethical Concerns:**

["NO or VERY MINOR ethics concerns only"]

**Final Justification:**

After reading the rebuttal, I think the work has important contribution to the theory explaining tradeoffs between learnability and expressivity. Moreover, the work sheds light into the role of convexity (in terms of both the loss landscape as well as the decision boundary) in such tradeoff. Therefore, I am now leaning towards accept.

**Limitations:**

Yes.

**Paper Formatting Concerns:**

No.

**Quality:**

3

**Strengths And Weaknesses:**

**Strengths**
This paper presents a technically solid and mathematically rigorous study of binary classification using starshaped polyhedral sets, contributing novel insights into the combinatorial and geometric structure of hypothesis class with non-convex decision boundaries.
The hypothesis class constructed is an insightful example of efficiently learnable classes with significance non-convex structure. The work excels in its formal analysis: it gives explicit bounds on VC-dimension, and characterizes the geometry of sublevel sets for both 0-1 and exponential loss functions.

**Weaknesses**
The paper seems to be only loosely connected to the broader machine learning community. It will be nice if the authors can discuss in more detail the connection to other common themes in machine learning such as the tension between expressivity and learnability, and the effect of the choice of loss function to efficient learnability.
Besides, the definitions appearing in the paper, e.g., polyhedral fan, simplifical fan, are quite complicated and dense. It will be nice if the authors can provide pointers to the formal definition at least during the first few occurences of these definitions. It may also be helpful to provide more visual examples to help non-expert readers parse these geometric concepts.

---

> ### Author Rebuttal · Authors · 2025-07-31
>
> We thank the reviewer for the careful examination of our manuscript and the thoughtful questions and comments. We hope to address the main concerns in our response below. To support our theoretical results, we additionally conducted a small-scale experimental study, which we describe below.
>
>
> > *It will be nice if the authors can discuss in more detail the connection to other common themes in machine learning such as the tension between expressivity and learnability, and the effect of the choice of loss function to efficient learnability.*
>
>
>  We appreciate the reviewer’s suggestion to more clearly situate our work within broader machine‐learning themes. In the revised manuscript, we emphasize the following points:
>
> - Our star‐shaped hypothesis class sits between convex piecewise‐linear (limited expressivity but easy to learn) and fully piecewise‐linear (very expressive but high complexity). By proving that the VC dimension equals the number of rays, we quantify precisely how increasing the number of linear regions of the classifier (and thus expressivity) impacts sample complexity, making explicit the classic trade-off between richer decision boundaries and learnability.
>
>
> - The effect of the choice of the loss function is made explicitly on two concrete examples:  0‑1 loss (whose sublevel sets decompose into unions of chambers of a hyperplane arrangement) with an exponential loss function (whose superlevel sets are concave). This contrast illustrates how the geometry induced by different losses governs optimization: while 0‑1 loss minimization is NP‑hard in general, the star‑convex structure admits principled local‑search heuristics, and the concavity of the exponential loss admits efficient convex solvers.
>
>
> > *The definitions appearing in the paper, e.g., polyhedral fan, simplifical fan, are quite complicated and dense. It will be nice if the authors can provide pointers to the formal definition at least during the first few occurences of these definitions. It may also be helpful to provide more visual examples to help non-expert readers parse these geometric concepts.*
>
>
> We regret that, due to strict page limits, our background section could not fully elaborate on the rich theory of polyhedral and simplicial fans.
> In the revised version we will add clearer pointers to the examples and visualizations in Appendix A, and add explicit pointers (e.g., Ziegler’s Lectures on Polytopes, Chapters 1–2) for additional material introducing the formal definitions at the first mention of "polyhedral fan" and "simplicial fan."
> If the reviewer has concrete suggestions for specific examples beyond the material in Appendix A, we would be delighted to incorporate them.
>
> > *Can you elaborate the significance of Corollary 3.3, Theorem 3.4 and Theorem 3.5?*
>
> Our geometric analysis reveals that sublevel sets under the 0-1 loss decompose into convex chambers within a hyperplane arrangement (Corollary 3.3), while simultaneously exhibiting star‑convexity with respect to any optimal point in the chamber which minimizes the loss (Th. 3.4 and 3.5).
> These theorems establish that while global optimization over the 0-1 loss is hard  due to the combinatorial complexity of the overall non‑convex landscape (Cor. 3.3), the local landscape is well-connected (Th. 3.4, Th. 3.5). In particular, for separable data these results show that local optimization methods can move from a chamber to a neighboring chamber along a straight line, improving the number of mistakes in each step and finding and terminating at the global minimum (without getting stuck at local minima (Th. 3.5). Theorem 3.4 says that also in the case of non-separable data, the landscape of false positives and false negatives behaves similarly. These results contrast with arbitrary non-convex losses, where disconnected sublevel sets can trap local methods, and highlights a key advantage of our structured non-convex setting.
>
> **Numerical Experiments:**
>
> We conducted a small-scale study using synthetic two-dimensional data, comparing our algorithm with standard binary classification models that also involve convex optimization, as well as simple neural network architectures. We focused on the case without translation. We plan to include these results into the camera-ready version.
>
> Preliminary results suggest that our algorithm achieves slightly better accuracy than SVM with a radial basis function kernel and comparable or slightly better performance than neural networks with ReLU activation. Logistic regression and SVM with polynomial kernels performed poorly on our star-convex data.
>
> We also investigated the impact of fan selection. Specifically, we tested data sampled from stars with fewer rays than the underlying fan of our model. The "dents" introduced by these new rays were, unsurprisingly, not very significant in magnitude. This observation supports the following heuristic in two dimensions: begin with a small number of rays and iteratively refine the fan by adding more rays. If additional rays do not yield significant new dents, they may be safely discarded.

---

> > ### Comment · Reviewer_cg74 · 2025-08-05
> > **Rebuttal Response**
> >
> > Thank you for the detailed response. I now have much better understanding of the significance of the result in the context of machine learning. The interpretation of Theorem 3.4 and Theorem 3.5 is also very helpful.

---

### Official Review · Reviewer_nJxy · 2025-06-27

**Clarity:** 4
**Significance:** 2
**Originality:** 4
**Rating:** 4
**Confidence:** 4

**Summary:**

This paper studies the combinatorial and geometric properties of binary classifiers whose decision boundaries are defined by star shape sets, i.e., and non-convex piecewise linear functions supported on a fixed polyhedral simplical fan of cones. The authors present bounds on the VC dimension, demonstrating learn ability, as well as other geometric/convexity/partial convexity properties of a zero-one loss function as well as an exponential type loss function. The further study related properties of an extension that allows for choosing a translation of the star-shaped set as well.

**Questions:**

1. If it is possible for the authors to include a numerical experiment in their camera ready version of the paper, which compares their proposed model to standard binary classification models that result in convex optimization problems, for example linear models, SVMs, logistic regression, kernel methods etc. this may help to improve my score. Critically the experiment needs to be comprehensive and sound enough, and include comparisons along the dimensions of performance of the models as well as computational difficulty. In my mind, an adequate experiment only needs to be performed on a simple tabular dataset(s), although comparisons with neural networks (which also yield nonlinear decision boundaries) might be interesting as well.

2. Why is the restriction $a > 0$ needed on the vectors defining the classifiers? Is it related to my comment above about the threshold being at 1 instead of 0? If you redefined the threshold to zero, could $a$ be unrestricted?

3. I disagree with the comment that the log likelihood function in equation (2) penalizes false positives and false negatives differently. As I mentioned, this is really a binary cross entropy type function, and it should be symmetric with respect to the two classes. Note that the reason for the different logarithmic versus linear forms of the two terms arises from the form of the exponential distribution, which does not favor one class more than the other. Furthermore, the comment on the different rates of decay between the two terms is not totally sound because $f_a$ is bounded below by zero but is unbounded above. To properly have a difference in how false positives and false negatives are treated, one we need to introduce a reweighting on the two terms. I would suggest revising this discussion.

4. Properly speaking, Problem 2 in algorithm 1 is not convex because the constraint $a > 0$ is an open set. If kept in this form, can it be made to be $a \geq 0$?

5. Minor:  definition 2.3 seems to be referring to a *proper* face.

**Ethical Concerns:**

["NO or VERY MINOR ethics concerns only"]

**Final Justification:**

I have adjusted my score to a 4. After reading the rebuttal and the other reviews, I think this is a nice contribution to learning theory and its interplay with polyhedral theory. However, my reason for borderline accept is that the experiments that the authors mentioned are only on very small synthetic datasets and therefore limited, in particular on the evaluation dimension of computational work/difficulty/scalability. Also, due to the limitations of the NeurIPS OpenReview platform, the authors are not able to share their experiment results. However, despite these issues the theoretical contribution of the paper is strong and so I lean towards acceptance. With strong numerical evidence, this paper would be a definite (and not borderline) accept for me.

**Limitations:**

yes

**Quality:**

3

**Strengths And Weaknesses:**

Strengths:
1. This is a theoretically solid paper, studying a family of binary classifiers that, the best of my knowledge, is novel and has the potential to be significant. I think this is an interesting and valuable family of classifiers, particularly because they sit in between convex piecewise linear and fully piecewise linear functions, as the authors describe. It is therefore very interesting to examine whether this non-convex yet somehow structured family of classifiers has desirable properties.
2. The authors present a thorough and comprehensive study of various geometric and convexity properties, which I appreciate.
3. The paper is written and organized, well and the results appear to be mathematically sound (although I did not read the appendix in full).
4. I particularly find the following two results to be most significant and useful for ML researchers. First, the result of Theorem 3.1, the characterization of the VC dimension of the set of star shaped classifiers for a fixed simplicial fan as exactly the number of generators of the fan, is very natural to me and insightful to understanding the complexity of learning. Second, Thm 3.6, which demonstrates concavity of the exponential type loss function, is very helpful for verifying the computational tractability of learning in this setting.

Weaknesses:
1. To me, a major weakness here is the lack of numerical experiments, which diminishes the (current) significance of the results. Well I understand that this is a theoretical paper and the authors mentioned this in their limitations discussion, I still think it is critical (and would expect) to include some basic numerical experiments in a paper like this. Demonstrating the value of this novel family of binary classification models, even on a very simple synthetic or real binary classification task (i.e., with tabular data) would help to clarify to the ML community that there is some practicality to the theoretical ideas of this paper. Particularly because the exponential loss leads to a convex optimization problem, in the setting of Thm 3.6, it is quite disappointing that there is no experiment to demonstrate the solvability of this convex problem and the relative merits of this family of classifier functions, which would help to further support and justify their choice over other models.
2. At the same time, the section and results on translations and and translated versions of the stars shaped sets, seem less impactful to me. Indeed, in neural net activation functions for example, these types of translations are often not considered, at least until the case without translations studied more carefully. I would've rather seen a more focused discussion on the case without translation, including experiments, that needs additional results that don't necessarily have as much value.
3. The authors make some choices in the development of their model, that in my opinion from the perspective of learning theory, are not totally standard. Most notably:
- The decision to set the threshold of the classifier for predicting positive versus negative to one, instead of to zero, which recovers the sign function and is by far the standard in binary classification (often referred to as "margin based" classification). It is not clear to me why the threshold needs to be set to one, and I think zero would be more standard and more elegant potentially.
- the "exponential loss function" proposed here is different from the standard exponential loss of binary classification, where the loss is $e^{-yf(x)}$ and is particularly notable in the boosting algorithm literature. In fact, the exponential loss considered in this paper is really much more like a cross entropy loss. I would suggest calling it that, or a version of that, instead of the exponential loss.

---

> ### Author Rebuttal · Authors · 2025-07-31
>
> We thank the reviewer for their careful examination of our manuscript and for the thoughtful questions and comments. We hope to address the main concerns in our response below, and also conducted a small-scale experimental study as suggested, see below.
>
> > *To me, a major weakness here is the lack of numerical experiments, which diminishes the (current) significance of the results.*
>
> We fully agree that empirical validation would greatly enhance the impact of our work. In the revised version of our article, we will include a small-scale study on two-dimensional data, which we now describe:
>
> We conducted a small-scale study using synthetic two-dimensional data, comparing our algorithm with standard binary classification models that also involve convex optimization, as well as simple neural network architectures. We focused on the case without translation.
>
> Preliminary results suggest that our algorithm achieves slightly better accuracy than SVM with a radial basis function kernel and comparable or slightly better performance than neural networks with ReLU activation. Logistic regression and SVM with polynomial kernels performed poorly on our star-convex data.
>
> We also investigated the impact of fan selection. Specifically, we tested data sampled from stars with fewer rays than the underlying fan of our model. The "dents" introduced by these new rays were, unsurprisingly, not very significant in magnitude. This observation supports the following heuristic for choosing an appropriate fan in two dimensions: begin with a small number of rays and iteratively refine the fan by adding more rays. If additional rays do not yield significant new dents, they may be safely discarded.
>
> > *The section and results on translations and and translated versions of the stars shaped sets, seem less impactful to me.*
>
> We agree that the core theoretical insights center around origin-fixed fans. For this reason, we have structured the article to emphasize the case without translations first (Section 2), followed by translation as an extension (Section 3). In the camera‑ready version, our new experimental section will focus exclusively on the non‑translated model,  emphasizing the significance of Section 2.
>
>
> > *The decision to set the threshold of the classifier for predicting positive versus negative to one, instead of to zero, which recovers the sign function and is by far the standard in binary classification (often referred to as "margin based" classification). It is not clear to me why the threshold needs to be set to one, and I think zero would be more standard and more elegant potentially.*
>
> The choice of threshold is without loss of generality due to positive scaling of $a$: any positive-, unit-threshold model (as in our article) can be equivalently represented by a model with threshold zero and unrestricted domain (i.e., the sign function) by applying the logarithm. The reason for our choice lies in the geometric construction and the general geometry of star bodies and their radial functions. We have chosen to keep the number of functions as minimal as possible, thus omitting the additional step of composing the classifier with a logarithmic function.
>
>
> > *The "exponential loss function" proposed here is different from the standard exponential loss of binary classification, where the loss is  and is particularly notable in the boosting algorithm literature. In fact, the exponential loss considered in this paper is really much more like a cross entropy loss. I would suggest calling it that, or a version of that, instead of the exponential loss.*
>
>
> We thank the reviewer for this remark. We will rename it to “log-likelihood loss”.
>
>
> > *I disagree with the comment that the log likelihood function in equation (2) penalizes false positives and false negatives differently.*
>
> Thank you very much for pointing this out, this was misconception on our side. We will revise the paragraph about the bias towards false positives and false negatives in the camera-ready version of the article.
>
> > *Why is the restriction $a>0$ needed on the vectors defining the classifiers? Is it related to my comment above about the threshold being at 1 instead of 0? If you redefined the threshold to zero, could  be unrestricted?*
>
> The restriction lies in the nature of model -- in fact, the choice of the threshold for the classifier being 1 comes from the restriction of the $a$'s to be positive, not the other way around. The entries of the vector $a$ correspond to positive scalars of the rays of the fan, which define the stars. A negative scalar would correspond to "flipping" the ray direction to its negative. The associated classifier would not be star-convex (and might even have holes), thus lying outside our class of star-convex classifiers.
>
> > *Properly speaking, Problem 2 in algorithm 1 is not convex because the constraint $a> 0$ is an open set. If kept in this form, can it be made to be $a \geq 0$?*
>
>  Indeed, this is a very good point. Strictly speaking, for it to be a convex program, we need to consider the closed positive orthant. Any solution on the boundary corresponds then to a degenerate star, where one point on a ray moves to infinity. This is considered in Theorem 3.9, and we will highlight this subtlety more clearly in the revised version.
>
> > *5. Minor: definition 2.3 seems to be referring to a proper face.*
>
> Thank you for pointing this out, we will correct this in the revised version.

---

> ### Comment · Reviewer_nJxy · 2025-08-05
> **Response to Rebuttal**
>
> I appreciate the authors' response to my concerns and questions. Thank you for providing detailed answers, which helped clarify several points for me. I am glad to see that you ran some preliminary experiments and that the results are coherent with your expectations, although I do remark that the (synthetic and very small scale) experiments are currently somewhat limited. I have no further questions at this time.
>
> EDIT:  I want to add that I have adjusted my score to a 4. After reading the rebuttal and the other reviews, I think this is a nice contribution to learning theory and its interplay with polyhedral theory. However, my reason for borderline accept is that the experiments that the authors mentioned are only on very small synthetic datasets and therefore limited, in particular on the evaluation dimension of computational work/difficulty/scalability. Also, due to the limitations of the NeurIPS OpenReview platform, the authors are not able to share their experiment results. However, despite these issues the theoretical contribution of the paper is strong and so I lean towards acceptance. With strong numerical evidence, this paper would be a definite (and not borderline) accept for me.

---

### Official Review · Reviewer_jMmD · 2025-07-02

**Clarity:** 4
**Significance:** 3
**Originality:** 3
**Rating:** 4
**Confidence:** 2

**Summary:**

This paper introduces and analyzes a novel class of binary classifiers whose decision boundaries are star-shaped polyhedral sets. The model is built upon a pre-defined simplicial polyhedral fan, and the shape of the star is determined by a vector of positive parameters,
. The authors provide a thorough theoretical investigation of this model from a geometric and combinatorial perspective.

Some of the key contributions that I could mention are:

Expressivity Analysis: The paper establishes that the Vapnik-Chervonenkis (VC) dimension of the classifier class (with a fixed origin) is equal to the number of generators in the underlying fan.

Loss Landscape Geometry: It characterizes the parameter space for two different loss functions. For the 0/1-loss, the sublevel sets are described as unions of chambers in a hyperplane arrangement, with the sublevel sets exhibiting star-convexity. For a proposed exponential loss function, the log-likelihood is proven to be concave, guaranteeing that local optima are global and enabling efficient optimization.

Extended Model: The framework is extended to allow for translations of the star's origin, which are treated as additional parameters. The paper analyzes the geometry of this larger parameter space, showing that the sublevel sets of the 0/1-loss become semi-algebraic and can be disconnected. It also provides an upper bound for the VC dimension of this more expressive classifier family.

The work is purely theoretical, providing a foundation for this class of classifiers.

**Questions:**

1. The framework relies on a given simplicial fan. While determining the optimal fan is out of scope, could you provide more intuition on the practical implications of this choice? For example, how does the complexity of the fan (e.g., number of generators n) relate to the model's sample complexity (via the VC dimension) and its ability to fit complex decision boundaries? Are there any data-dependent heuristics one might consider for fan selection in a practical setting?

2. The introduction compellingly positions this work as an alternative to general ReLU networks. Could you elaborate on this connection? Clarifying this could make the work's relationship to the broader deep learning literature more explicit.

**Ethical Concerns:**

["NO or VERY MINOR ethics concerns only"]

**Final Justification:**

My concerns are addressed and I shall maintain the score.

**Limitations:**

I included some questions in the weaknesses and questions section. It would be great to hear responses on those.

**Quality:**

3

**Strengths And Weaknesses:**

*Strengths*

1. The paper proposes a novel and interesting family of classifiers that elegantly bridges the gap between simple linear/convex models and highly complex, non-convex models like general ReLU neural networks. By fixing the underlying combinatorial structure (the fan), the authors introduce a controllable and analyzable form of non-convexity. This is a strong and original theoretical contribution to the geometric understanding of machine learning models.

2. Although not an expert in this area, I like the technical quality of this work. The analysis skillfully combines concepts from polyhedral geometry, convex analysis, and statistical learning theory. The characterization of the loss landscapes, particularly the concavity result for the exponential loss is very interesting.

3. The paper is exceptionally well-written and organized. The authors take care to define all necessary geometric concepts before using them.


*Weakness*
The primary weakness, as acknowledged by the authors, is the absence of any experiments. While this is a theoretical paper, the ultimate motivation is classification. Even a small-scale experimental study on synthetic data could have provided valuable insights.

---

> ### Author Rebuttal · Authors · 2025-07-31
>
> We thank the reviewer for their careful examination of our manuscript and for the thoughtful questions and comments. We aim to address the main concerns in our response below, and also conducted a small-scale experimental study as suggested, see below.
>
> > *The primary weakness, as acknowledged by the authors, is the absence of any experiments. While this is a theoretical paper, the ultimate motivation is classification. Even a small-scale experimental study on synthetic data could have provided valuable insights.*
>
> As suggested by the reviewer, we conducted a small-scale study using synthetic two-dimensional data, comparing our algorithm with standard binary classification models that also involve convex optimization, as well as simple neural network architectures. We focused on the case without translation. We plan to include these results into the camera-ready version.
>
> Preliminary results suggest that our algorithm achieves slightly better accuracy than SVM with a radial basis function kernel and comparable or slightly better performance than neural networks with ReLU activation. Logistic regression and SVM with polynomial kernels performed poorly on our star-convex data.
>
> We also investigated the impact of fan selection. Specifically, we tested data sampled from stars with fewer rays than the underlying fan of our model. The "dents" introduced by these new rays were, unsurprisingly, not very significant in magnitude. This observation supports the following heuristic in two dimensions: begin with a small number of rays and iteratively refine the fan by adding more rays. If additional rays do not yield significant new dents, they may be safely discarded.
>
> > *The framework relies on a given simplicial fan. While determining the optimal fan is out of scope, could you provide more intuition on the practical implications of this choice? For example, how does the complexity of the fan (e.g., number of generators n) relate to the model's sample complexity (via the VC dimension) and its ability to fit complex decision boundaries? Are there any data-dependent heuristics one might consider for fan selection in a practical setting?*
>
> The number of rays has a direct impact on the model's complexity: by a result of Stanley (1975), the number of maximal cones (and thus linear pieces in the decision boundary) in a simplicial fan with n rays is $n^{d/2}$. Thus, given a model with n rays, the models complexity is bounded by $n^{d/2}$. In practical terms this means that adding more rays gives you finer decision boundaries, but also raises the sample complexity and computational burden.
>
> The choice of an appropriate fan is indeed crucial and nontrivial. As a data-dependent heuristic in two dimensions, we suggest to incrementally increase the number of rays; see the description of our experiments above.  This strategy allows the fan to grow adaptively with observed structure, avoiding over-parameterization while still capturing complex decision regions.
>
>
> > *The introduction compellingly positions this work as an alternative to general ReLU networks. Could you elaborate on this connection? Clarifying this could make the work's relationship to the broader deep learning literature more explicit.*
>
>
> Our family of star-shaped classifiers falls within the broader class of continuous piecewise-linear functions, and as such can be represented by a suitably structured ReLU network. However, the powerful flexibility of ReLU networks makes it challenging to enforce specific geometric properties, such as ensuring that the decision region satisfies star-convexity. In contrast, by building our classifiers directly on a fixed simplicial fan, we retain the ability to model non-convex boundaries yet maintain high control over the shape and connectivity of the decision regions. In other words, whereas a generic ReLU network may approximate virtually any decision boundary but offers little transparency or guarantees on its geometry, our approach imposes a disciplined piecewise-linear structure that both captures rich non-convex phenomena and admits a precise theoretical analysis.

---

### Official Review · Reviewer_EcQ6 · 2025-07-17

**Clarity:** 4
**Significance:** 1
**Originality:** 2
**Rating:** 3
**Confidence:** 3

**Summary:**

The authors consider the optimization of star-shaped sets. To do so, they are given a set of simplicial cones which are a set of cones than span $\mathbb{R}^d} whose intersection is itself a facet of the cones). As a result, for every point $x$ one can decompose it as a combination of the basis vectors for some cone denoted $[x]$. Assuming that such a simplicial fan has $n$ vectors in $\mathbb{R}^d$, we can write each vector  $x$   as  a $d$-sparse combination fo the $n$ basis vectors. For a given $a \in \mathbb{R}_{>0}^n$, we can define the hyperplane arangement that defines a set $[x]^T\cdot a$ and try to learn this separation.

This motivates their first result which is that the VC dimension is $n$. Next they show that the sublevel sets for the discrete loss are star-convex and for the exponential loss, superlevel sets are concave. Lastly they consider the problem where both the offset $t$ and the parameter $a$. In this case the set of losses are semi-algebraic sets.

**Questions:**

Can you elaborate on how the appropriate simplicial fans will be chosen. Why this method would be better than other existing techniques?

**Ethical Concerns:**

["NO or VERY MINOR ethics concerns only"]

**Final Justification:**

I thank the authors for their response. I appreciate their contributions and the work they have put in. However, it doesn't quite fit the bar for acceptance in my books. I feel adding some of the additional directions they mentioned in their response will make a much stronger paper.

**Limitations:**

Yes

**Quality:**

2

**Strengths And Weaknesses:**

The paper is complete and has a thorogh set of results.

The weakeness is that the technical novelty of the paper is limited. None of the theorems or proofs are surprising and are fairly natural. I think showing that such simplicial fans are useful either theoretically for some class of problems or for some empirical datasets would make the paper much stronger. It would be useful to share any mechanisms  on how to choose appropriate simplicial fans is important.

---

> ### Author Rebuttal · Authors · 2025-07-31
>
> We thank the reviewer for their careful examination of our manuscript and for the thoughtful questions and comments. We aim to address the main concerns in our response below.
>
> We acknowledge the reviewer’s concern regarding the novelty of the individual technical results. While the mathematical tools we employ are classical, our primary contribution lies in formally establishing connections between simplicial fans and learning theory. Specifically, we analyze the VC dimension, star-convexity of discrete losses, and the concavity properties of exponential losses. To the best of our knowledge, this systematic exploration is novel within the context of optimizing star-shaped sets and contributes to a foundational understanding that can support further theoretical and practical developments.
>
> To better support the significance of our results, we plan to expand the current version of the article with a small-scale computational study on synthetic data. This study compares our algorithm with standard approaches such as SVM, logistic regression, and neural networks; a description of our preliminary results is provided below.
>
> The choice of an appropriate fan is indeed crucial and nontrivial. As a data-dependent heuristic in two dimensions, we suggest to incrementally increase the number of rays; see the description of our experiments below.  This strategy allows the fan to grow adaptively with observed structure, avoiding over-parameterization while still capturing complex decision regions.
>
>
> **Numerical Experiments:**
>
> We conducted a small-scale study using synthetic two-dimensional data, comparing our algorithm with standard binary classification models that also involve convex optimization, as well as simple neural network architectures. We focused on the case without translation.
>
> Preliminary results suggest that our algorithm achieves slightly better accuracy than SVM with a radial basis function kernel and comparable or slightly better performance than neural networks with ReLU activation. Logistic regression and SVM with polynomial kernels performed poorly on our star-convex data.
>
> We also investigated the impact of fan selection. Specifically, we tested data sampled from stars with fewer rays than the underlying fan of our model. The "dents" introduced by these new rays were, unsurprisingly, not very significant in magnitude. This observation supports the following heuristic in two dimensions: begin with a small number of rays and iteratively refine the fan by adding more rays. If additional rays do not yield significant new dents, they may be safely discarded.

---

> > ### Author Response · Authors · 2025-08-09
> >
> > We would kindly like to check whether our responses addressed your concerns, or if there are any points that remain unclear and would benefit from further clarification.

---

### Decision · Program_Chairs · 2025-09-17

**Decision:**

Accept (poster)

**Comment:**

This paper presents a nice theoretical study of binary classification with star-shaped polyhedral decision boundaries, establishing VC dim bounds and characterizing the geometry of loss landscapes. The results are technically solid and novel in framing, though largely based on classical tools. Some weakenesses were identified in the rebuttal phase, which the authors addressed with preliminary experiments and a clarification of the broader significance.

The authors should take the excellent points in the discussion into account as they craft a final version of the manuscript. Overall, this is a valuable contribution.